



Measurement report: Fireworks impacts on air quality in Metro Manila, Philippines during the
2019 New Year revelry
Genevieve Rose Lorenzo[1,2], Paola Angela Bañaga[2,3], Maria Obiminda Cambaliza[2,3], Melliza
Templonuevo Cruz[3,4], Mojtaba AzadiAghdam[6], Avelino Arellano[1], Grace Betito[3], Rachel
Braun[6], Andrea F. Corral[6], Hossein Dadashazar[6], Eva-Lou Edwards[6], Edwin Eloranta[5], Robert
Holz[5], Gabrielle Leung[2], Lin Ma[6], Alexander B. MacDonald[6], James Bernard Simpas[2,3], Connor
Stahl[6], Shane Marie Visaga[2,3],  Armin Sorooshian[1,6]
[1]Department of Hydrology and Atmospheric Sciences, University of Arizona, Tucson, Arizona,
85721, USA
[2]Manila Observatory, Quezon City, 1108, Philippines
[3]Department of Physics, School of Science and Engineering, Ateneo de Manila University,
Quezon City, 1108, Philippines
[4]Institute of Environmental Science and Meteorology, University of the Philippines, Diliman,
Quezon City, 1101, Philippines
[5]Space Science and Engineering Center, University of Wisconsin - Madison, Madison,
Wisconsin, 53706, USA
[6]Department of Chemical and Environmental Engineering, University of Arizona, Tucson,
Arizona, 85721, USA
*Correspondence to: armin@email.arizona.edu*





**Abstract**
Fireworks degrade air quality, reduce visibility, alter atmospheric chemistry, and cause short-
term adverse health effects. However, there have not been any comprehensive physicochemical
and optical measurements of fireworks and their associated impacts in a Southeast Asia
megacity, where fireworks are a regular part of the culture. Size-resolved particulate matter (PM)
measurements were made before, during, and after New Year 2019 at the Manila Observatory in
Quezon City, Philippines, as part of the Cloud, Aerosol, and Monsoon Processes Philippines
Experiment (CAMP$^2$Ex). A High Spectral Resolution Lidar (HSRL) recorded a substantial
increase in backscattered signal associated with high aerosol loading ~440 m above the surface
during the peak of firework activities around 00:00 (local time). This was accompanied by PM$_{2.5}$
concentrations peaking at 383.9 $\mu$g m$^{-3}$. During the firework event, water-soluble ions and
elements, which affect particle formation, growth, and fate, were mostly in the submicrometer
diameter range. Total (> 0.056 $\mu$m) water-soluble bulk particle mass concentrations were
enriched by 5.7 times during the fireworks relative to the background (i.e., average of before and
after the firework). The water-soluble mass fraction of PM$_{2.5}$ increased by 18.5% above that of
background values. Bulk particle hygroscopicity, kappa ($\kappa$), increased from 0.11 (background) to
0.18 (fireworks). Potassium and non-sea salt (nss) SO$_4^{2-}$ contributed the most (70.9%) to the
water-soluble mass, with their mass size distributions shifting from a smaller to a larger
submicrometer mode during the firework event. On the other hand, mass size distributions for
NO$_3^-$, Cl$^-$, and Mg$^{2+}$ (21.1% mass contribution) shifted from a supermicrometer mode to a
submicrometer mode. Being both uninfluenced by secondary aerosol formation and constituents
of firework materials, a subset of species were identified as the best firework tracer species (Cu,
Ba, Sr, K$^+$, Al, and Pb). Although these species (excluding K$^+$) only contributed 2.1% of the total
mass concentration of water-soluble ions and elements, they exhibited the highest enrichments
(6.1 to 65.2) during the fireworks. Surface microscopy analysis confirmed the presence of
potassium/chloride-rich cubic particles along with capsule-shaped particles in firework samples.
The results of this study highlight how firework emissions change the physicochemical and
optical properties of water-soluble particles (e.g., mass size distribution, composition,
hygroscopicity, and aerosol backscatter), which subsequently alters the background aerosol's
respirability, influence on surroundings, ability to uptake gases, and viability as cloud
condensation nuclei (CCN).


## 1. Introduction

Fireworks affect local populations through visibility reduction and increased health risks due to briefly elevated particulate matter (PM) levels. Total PM mass concentrations during local celebrations in different cities have reached up to 235 μg m$^{-3}$ (Leipzig, Germany) (Wehner et al., 2000), 700 μg m$^{-3}$ (Texas, United States [U.S.]) (Karnae, 2005), 1,510 μg m$^{-3}$ (Montreal, Canada) (Joly et al., 2010), and 2,582 μg m$^{-3}$ (New Delhi, India) (Mönkkönen et al., 2004). These levels exceed the 24 h U.S. National Ambient Air Quality Standard (NAAQS) for PM$_{10}$ of 150 μg m$^{-3}$. Firework emissions from at least nineteen studies have also been linked to exceedance of the 24 h U.S. NAAQS limit for PM$_{2.5}$ of 35 μg m$^{-3}$ (Lin, 2016 and references therein). Higher PM concentrations from fireworks have been reported more frequently in Asia (i.e., India, China, and Taiwan) compared to Western countries (Lin, 2016; Sarkar et al., 2010).

Health effects are of major concern during firework periods based on both short and long-term exposure. For example, Diwali is a major firework festival in India where it was shown that chronic exposure to three of the most prominent tracer species (Sr, K, and Ba) translated to a 2% increase in health effects based on the non-carcinogenic hazard index (Sarkar et al., 2010). On the other hand, short term exposure to firework pollutants increases asthma risk, eye allergies, cardiovascular and pulmonary issues, cough, and fever (Moreno et al., 2010; Singh et al., 2019; Barman et al., 2008; Becker et al., 2000; Beig et al., 2013; Hirai et al., 2000). Firework pollutants also impact clouds and the hydrological cycle, owing to associated aerosols serving as cloud condensation nuclei (CCN) (Drewnick et al., 2006) and subsequently impacting surface ecosystems after wet deposition (Wilkin et al., 2007). Although fireworks emit particles with various sizes into the atmosphere, fine particles associated with PM$_{2.5}$ are most relevant to public health effects, scattering efficiency, and CCN activation (Vecchi et al., 2008; Perry, 1999). Knowing the various effects of firework emissions depends on knowing their physical, chemical, and optical properties.

Measurements of the chemical composition of firework emissions are important in order to understand how they affect local air quality. The main components of fireworks are fuels (metals and alloys, metalloids, and non-metals), oxidizers (nitrates, perchlorates, and chlorates), and coloring agents (metal salts) (Steinhauser and Klapotke, 2010). Previous studies have relied on tracer species to establish confidence in distinguishing the firework source from background air and other sources (Sarkar et al., 2010). Potassium historically has been the most observable tracer for fireworks emissions (Wang et al., 2007; Drewnick et al., 2006; Perry, 1999), with concentrations reaching 58 μg m$^{-3}$ during the Diwali Festival in India (Kulshrestha et al., 2004). Firework color is created by metal salts such as Sr for red, Ba for green, and Cu for blue, all three of which have and have been found to be effective tracers of fireworks (Walsh et al., 2009; Vecchi et al., 2008). Strontium in particular is an indicator of the spatial and temporal extent of firework smoke plumes (Perry, 1999) because of the high prevalence of red in fireworks and it is not affected by traffic emissions (Moreno et al., 2010). Other components measured in the air that have been attributed to fireworks include metals such as Al, Cd, Cu, Ti, Mg, Mn, Ni, Zn, their salt anion counterparts (S, P, Cl) and other trace metals (As, Bi, Co, Ga, Hg, Cr, Pb, Rb, Sb, and P). Also from fuel and oxidizer combustion are species such as NO$_3^-$, SO$_4^{2-}$, and organics including oxaloacetic acid (Alpert and Hopke, 1981; Barman et al., 2008; Carranza et al., 2001;



Dorado et al., 2001; Drewnick et al., 2006; Joly et al., 2010; Joshi et al., 2016; Kulshrestha et al.,
2004; Kumar et al., 2016; Lin et al., 2016; Moreno et al., 2010; Sarkar et al., 2010; Tanda et al.,
2019; Thakur et al., 2010; Joshi et al., 2019). Black carbon mass concentrations during firework
events can either increase due to firework emissions or decrease owing to fewer vehicles on the
road (Kumar et al., 2016; Yadav et al., 2019). In both cases, the black carbon mass fraction
decreases due to a greater contribution of other constituents in firework emissions. Organic mass
concentrations and mass fractions have been noted to increase and decrease, respectively, with
fireworks (Zhang et al., 2019). Governed largely by composition, particulate hygroscopicity and
solubility have also been found to be altered by fireworks depending on the emitted species.
Copper and Mg were observed to become more soluble in firework emissions in Delhi, India,
while Mn, As, Ba, and Pb became less soluble (Perrino et al., 2011). The water-soluble aerosol
component from fireworks in Sichuan Basin (China) were internally mixed and enhanced the
hygroscopicity of submicrometer aerosols, especially the larger particles (Yuan et al., 2020).
In addition to composition, a necessary aspect of characterizing impacts of firework emissions is
to measure aerosol size distributions within the short timeframe of an event (Joshi et al., 2019).
Owing to combustion during firework events, PM concentrations are dominated by particles in
the submicrometer range (Vecchi et al., 2008; Nicolás et al., 2009; Joshi et al., 2019; Pirker et
al., 2020; Do et al., 2012). Particle number concentration maxima have been noted for the
nucleation (0.01 to 0.02 µm) and Aitken (0.02 to 0.05 µm) modes (Yadav et al., 2019; Yuan et
al., 2020), in addition to both the small (0.1 to 0.5 µm) (Wehner et al., 2000; Zhang et al., 2010)
and large (0.5 to 1.0 µm) ends of the accumulation mode (Vecchi et al., 2008) during firework
events. There are a few studies with observed particle mass concentration increases in the coarser
but still respirable (< 10 µm) mode (Tsai et al., 2011). In terms of dynamic behavior in the size
distributions, past work has shown a shift in number concentration from nucleation and Aitken
modes to the smaller end of the accumulation mode (0.1 to 0.5 µm), due to increased coagulation
sinks (Zhang et al., 2010). Finer temporal scale monitoring has revealed steep increases in
nucleation mode and Aitken mode particle concentrations associated with firework emissions
followed by a growth in accumulation mode particle number concentrations due to coagulation
(Yadav et al., 2019). An opposite shift to a smaller size distribution has been observed for certain
species (Mg, Al, Cu, Sr, and Ba) from the coarse mode to accumulation mode (Tanda et al.,
2019). Other work has shown that while there is usually a quick drop in particle concentration to
background values after firework events (Joly et al., 2010), elevated number concentrations of
accumulation mode particles are maintained for up to three hours after peak firework activity
(Hussein et al., 2005). New particle formation events with fireworks have also been reported in
Mumbai, India (Joshi et al., 2016), with enrichments of primary and secondary particles for up to
30 minutes after peak firework activity. Particle aging due to distance from the source and
meteorology alter firework emission particle concentrations (Joly et al., 2010) and size
distributions (Khaparde et al., 2012).
Meteorological and dynamic parameters such as wind speed, level of mixing (turbulent kinetic
energy), and mixing layer height (Lai and Brimblecombe, 2020) influence peak concentration
and composition of aerosols after fireworks, as well as particle residence time in the atmosphere





and transport to nearby regions (Vecchi et al., 2008). Although firework activities are episodic,
their particulate emissions, especially in the submicrometer mode (Do et al., 2012), reside in the
atmosphere for as long as several days to weeks (Liu et al., 1997; Lin et al., 2016; Kong et al.,
2015; Do et al., 2012). Dispersion of the particles under low wind speed (1 m s$^{-1}$) for particles
between 0.4 and 1 µm is estimated at 12 h (Vecchi et al., 2008) and can reach distances as far as
a hundred kilometers (Perry, 1999). Aitken mode and larger particles are dispersed by wind more
than nucleation-mode particles (Agus et al., 2008). Meteorological conditions, such as rainfall,
can also decrease firework particle loading in the air and relative humidity can change the
hygroscopicity of firework emissions (Hussein et al., 2005), thereby affecting their size and
radiative properties.
There currently is no in-depth analysis of the chemical, physical, and optical properties of
firework emissions in a Southeast Asian megacity where fireworks are culturally significant. This
work reports on size-resolved aerosol characteristics during the 2019 New Year celebrations in
Metro Manila, Philippines, one of the most populated cities, with 12.88 M population (PSA,
2015), in Southeast Asia. We address the following questions in order: (i) what is the
meteorological backdrop during the study period in relation to PM$_{2.5}$ levels; (ii) what is the effect
of the firework emissions on optical properties of aerosols?; (iii) what are the concentrations and
mass size distribution characteristics of different elemental and ionic species?; (iv) what are the
most enhanced tracers in firework emissions?; (v) what are the size-resolved morphological
characteristics of firework aerosols?; (vi) how does aerosol hygroscopicity respond to firework
emissions? The results of this work provide new data that can help address how past and on-
going firework emissions impact health, visibility, regional air quality, and biogeochemical
cycling of nutrients and contaminants in the Philippines, Southeast Asia, and, more broadly, for
all other cities with major firework events. It also contributes to the growing body of firework
research findings (Devara et al., 2015), with a unique feature of this work being the combination
of multiple data products, including surface-based lidar retrievals and size-resolved composition
and morphology analyses. Firework events are widespread episodes that can also be used to
expose and ultimately resolve differences between satellite and surface data (Williams et al.,
2005; Kumar et al., 2016).

**2. Methods**
2.1 Hourly PM$_{2.5}$ Mass Concentration
Hourly PM$_{2.5}$ mass concentrations were obtained to assess temporal characteristics of fine
particulates due to fireworks and their relation to meteorology and aerosol optical properties. The
hourly PM$_{2.5}$ mass concentrations were collected at the Manila Observatory, Quezon City,
Philippines (14.64° N, 121.08° E, ~70 m. a. s. l.) (Fig. S1) with a beta attenuation monitor
(DKK-TOA Corporation) as part of the East Asia Acid Deposition Monitoring Network
(EANET) (Totsuka et al., 2005). The beta attenuation monitor collects PM$_{2.5}$ samples on a ribbon
filter, which are irradiated with beta particles. The attenuation of the beta particles through the
sample and the filter is exponentially proportional to the mass loading on the filter. These hourly





data were then averaged over 48-hour periods coinciding with water-soluble aerosol composition
measurements (Section 2.4) before, during, and after the firework event.

2.2 Meteorological Data
Rainfall, temperature, relative humidity, and wind data were collected at the Manila Observatory
with a Davis Vantage Pro2 Plus weather station (~90 m. a. s. l) before, during, and after the
firework period. Hourly precipitation accumulation and 10-min averaged temperature, relative
humidity, and wind were used for the analysis.

2.3 Remote Sensing
Vertical profiles of aerosol backscatter cross-section measured with the University of Wisconsin
High Spectral Resolution Lidar (HSRL) which was deployed at the Manila Observatory in
support of CAMP$^2$EX. The HSRL instrument transmitting laser (Table S1) operates at 532 nm
with 250 mW average power and pulse repetition rate of 4 KHz. The HSRL technique measures
and separates the returned signal into the molecular and aerosol backscatter by using a beam
splitter and an iodine absorption cell filter. The separated molecular signal allows for optical
depth and backscatter cross section measurements in contrast to a standard backscatter lidar that
requires  assumption related to the particulate lidar ratio (Razenkov, 2010). The HSRL also
measures particulate depolarization ratio, an indicator of aerosol or cloud particle shape with low
depolarization indicative of spherical particles while intermediate values (10%) indicate a mix of
spherical and nonspherical particles (Burton et al., 2014; Reid et al., 2017).  HSRL data were
uploaded and processed at the University of Wisconsin-Madison Space Science and Engineering
Center server for periods before, during, and after the fireworks.

2.4 Aerosol Composition and Morphology Measurements
Size-speciated PM (cut-point diameters: 18, 10, 5.6, 3.2, 1.8, 1.0, 0.56, 0.32, 0.18, 0.10, and
0.056 μm) was collected on Teflon substrates (PTFE membrane, 2 μm pores, 46.2 mm diameter,
Whatman) with two Micro-Orifice Uniform Deposition Impactor (MOUDI II 120R, MSP
Corporation) (Marple et al., 2014) samplers from the third floor of the main building (~85 m. a.
s. l) at the Manila Observatory. Sample collection for each of the three MOUDI sets lasted 48
hours before (13:30 December 24, 2018 to 13:30 December 26, 2018), during (14:45 December
31, 2018 to 14:45 January 2, 2019), and after (13:30 January 1, 2019 to 13:30 January 3, 2019)
firework activities. Note all times refer to local time (UT + 8 hours). Although there were no
fireworks released from the sampling site, there was firework activity in the immediate vicinity
(~ 500 m from the sampling in all directions and all throughout the city in general). Firework
activity around the sampling site began around ~19:00 on December 31, 2018 and peaked at
00:00 of 1 January 2019).  There was limited firework after midnight. MOUDI samples collected
before and after the firework event were considered as background samples. Although there is




some firework activity that is expected in the evening of December 24 (before the firework
event), this is minimal compared to that which is the focus of this study. The samples were
covered with aluminum foil, sealed, and stored in the freezer before being shipped to the
University of Arizona for elemental and ionic analysis.
Each sample substrate was cut in half. One half of each sample was extracted in 8 mL Milli-Q
water (18.2 M$\Omega$cm), sonicated, and analyzed for ions (ion chromatography (IC): Thermo
Scientific Dionex ICS-2100 system) and elements (triple quadrupole inductively coupled plasma
mass spectrometer: ICP-QQQ; Agilent 8800 Series). The remaining substrate halves were stored.
Sample ionic and elemental concentrations were corrected by subtracting concentrations from
background control samples. More information about the sampling and analysis are detailed in
recent work (Stahl et al., 2020b). Limits of detection of the forty-one reported species are
summarized in Table S3. Potassium ($K^+$) was reported based on ICP-QQQ measurements rather
than IC due to possible contamination from the KOH eluent used in the latter instrument. Non-
sea salt $SO_4^{2-}$ was calculated by subtracting 0.2517 * $Na^+$ from the total $SO_4^{2-}$ concentration
(Prospero et al., 2003).
High-resolution scanning electron microscopy (SEM) combined with energy dispersive X-ray
analysis (EDX) was used for examining particle morphology and chemical composition on a
portion of the substrates collected during the firework event. Analyses were performed with a
Hitachi S-4800 high-resolution SEM and a Thermo Fisher Scientific Noran Six X-ray
Microanalysis System in the Kuiper Imaging cores at the University of Arizona. Approximately
1 cm$^2$ was cut from the center of substrate halves and placed on double-sided carbon tape
mounted on an aluminum stub. A thin layer (1.38 nm) of carbon was coated on the sample
surface using a Leica EM ACE600 sputter coater to improve the sample's conductivity. SEM
images were obtained at 15 keV and 30 keV acceleration voltages and with a 20 µA probe
current in high-magnification mode. The percentage contributions and the spatial distribution of
the elements were obtained from the EDX analysis. Carbon, F, and Al should be ignored in the
discussion of SEM-EDX results since C and F are present in the Teflon substrates, and the
sample stub is an Al-rich substrate.
2.5 Enrichment Factor Calculations
To identify which species are most enhanced during fireworks, enrichment values are typically
calculated using speciated concentrations during the fireworks relative to baseline periods
(Tanda et al., 2019). We calculate water-soluble mass enrichment factors for each of the forty-
one measured species by dividing their total bulk ($\geq 0.056\ \mu m$) mass concentrations during the
firework event by the average of the total mass concentration of the species measured before and
after the firework event. Size-resolved enrichments were similarly calculated using measured
mass concentrations for individual MOUDI stages. In a case when the mass concentration of a
species during the firework event was non-zero but the mass concentrations during and after
were zero (e.g., succinate), half of the detection limit was used in place of zero values.


2.6 Hygroscopicity Calculations
Hygroscopicity was calculated for particles ranging in size between 0.056 – 3.2 μm before,
during, and after the firework event. This size range was chosen to most closely be aligned with
separate measurements of PM$_{2.5}$ in the study (Section 2.1) that were used to account for the
remaining mass not speciated in this study. We specifically calculate values for the single
hygroscopicity parameter kappa, κ (Petters and Kreidenweis, 2007).
The water-soluble compound mass concentrations before, during, and after the firework event
were calculated using an ion-pairing scheme (Gysel et al., 2007) for each MOUDI stage between
diameters of 0.056 and 3.2 μm, and then summed to achieve a total mass concentration for each
compound in this size range. Black carbon mass concentrations in PM$_{2.5}$ before and after the
firework event were calculated based on their long-term (2001-2007) average contribution (32%)
to PM$_{2.5}$ mass in December and January (Cohen et al., 2009). Black carbon or elemental carbon
(EC) concentrations during the firework event were assumed to be the average of the black
carbon concentrations before and after the firework event. This was done because black carbon
concentrations have been observed to not increase (Santos et al., 2007) as much as organic
carbon (OC) (Lin, 2016), such that OC:EC mass ratios during fireworks have been observed to
increase. Total non-water-soluble content between 0.056 and 3.2 μm was calculated as the
difference between the total PM$_{2.5}$ mass concentration and the sum of the water-soluble species
and black carbon mass concentrations. The mass of each species was divided by its density, and
each of these volumes were added to quantify the volume of the measured aerosol (water-soluble
compounds, black carbon, and organic matter) between 0.056 and 3.2 μm. Volume fractions
were then computed for each species. The Zdanovskii, Stokes, and Robinson (ZSR) mixing rule
(Stokes and Robinson, 1966) was used to obtain the total hygroscopicity (total κ ) of the mixed
aerosols by weighting κ values for the individual non-interacting compounds by their respective
volume fractions and summing linearly. Densities and κ values for the individual compounds are
based on those used elsewhere (AzadiAghdam et al., 2019), repeated in Table S4.

2.7 Back Trajectories
Three-day back trajectories with six-hour resolution were generated using the National Oceanic
and Atmospheric Administration's (NOAA) Hybrid Single-Particle Lagrangian Integrated
Trajectory (HYSPLIT) model (Rolph et al., 2017; Stein et al., 2015) using the Global Data
Assimilation System (GDAS) with a resolution of 1°, and vertical wind setting of "model vertical
velocity". Back trajectories were chosen to end at the beginning times of the sampling periods
before, during, and after the firework event. Trajectories were computed for an end point being at
the Manila Observatory at an altitude of 500 m because it represents the mixed layer as done in
other works examining surface air quality (Mora et al., 2017; Aldhaif et al., 2020; Crosbie et al.,
2014; Schlosser et al., 2017).

**3. Results and Discussion**




3.1 Hourly PM$_{2.5}$, Meteorological, and Transport Patterns
Temporal analysis of PM$_{2.5}$ and meteorology (Fig. 1) can help in understanding how the
enhanced particulate concentrations detected at the Manila Observatory during the fireworks
evolved and were influenced by meteorology. Hourly PM$_{2.5}$ began to increase from 44.0 μg m$^{-3}$
(shortly after rising above the 24-h Philippine National Ambient Air Quality Guideline Value
(NAAQGV) of 50.0 μg m$^{-3}$) after 18:00 on 31 December 2018 with the beginning of firework
activity and calm meteorological conditions. There was moderate (3 mm) rainfall from 22:00 to
23:00 that night as the firework activity began to increase. Rain is a sink for particles (Perry,
1999) and could have washed out some of the particulates in the air, thus potentially causing a
slight dip in the hourly PM$_{2.5}$ around midnight. PM$_{2.5}$ peaked at 383.9 μg m$^{-3}$ between 01:00 to
02:00 on 1 January 2019. The PM$_{2.5}$ peak was delayed by approximately an hour from the peak
firework activity at midnight possibly due to rainfall, relative humidity, and wind (Vecchi et al.,
2008), in addition to aerosol dynamical processes requiring time for secondary aerosol formation
and growth (Li et al., 2017). Minimal rain (0.2 mm in an hour) with high relative humidity
(between 93% ± 4% to 94% ± 4%) were conducive to aerosol growth and/or secondary particle
formation. High relative humidity is related to aqueous-phase oxidation of SO$_2$ (Sun et al., 2013)
and NO$_2$ (Cheng et al., 2014) as well as metal-catalyzed heterogeneous reactions (Wang et al.,
2007) to form SO$_4^{2-}$. Aqueous oxidation has been found to be a predominant mechanism for the
secondary formation of SO$_4^{2-}$ during fireworks (Li et al., 2017), in addition to promoting
secondary organic aerosol formation (Wonaschuetz et al., 2012; Youn et al., 2013). Light wind
(~1 m s$^{-1}$) after midnight from the northeast could also have transported more emissions from the
populated Marikina Valley, located in the northeast, to the Manila Observatory contributing to
the delay of the PM$_{2.5}$ peak.
Particulate levels were enhanced for approximately 14 h from the beginning of the firework
activity (Fig. 1) during which the average PM$_{2.5}$ (143.4 μg m$^{-3}$) exceeded the 24 h Philippine
NAAQGV between 18:00 on 31 December 2018 to 08:00 on 1 January 2019. After 02:00 on 1
January 2019, PM$_{2.5}$ dropped quickly to 122.0 μg m$^{-3}$ between 03:00 to 04:00 (Fig. 1). The PM$_{2.5}$
decrease was less pronounced after 04:00 but continued decreasing steadily along with slight rain
(0.4 mm in an hour) and light breeze (1 – 2 m s$^{-1}$) from the northwest to southwest directions.
Firework activity in other countries have been documented to last from 2 – 6 h in a day and
elevated particulate levels can be maintained for up to 6 – 18 h (Thakur et al., 2010; Crespo et
al., 2012; Chatterjee et al., 2013; Kong et al., 2015; Tsai et al., 2012). The 48-h average PM$_{2.5}$
during (49.9 μg m$^{-3}$) the firework event was 1.9 and 3.3 times more, respectively, than before
(25.8 μg m$^{-3}$) (Fig. S2) and after (15.2 μg m$^{-3}$) (Fig. S3) the firework event. Previous work in
other countries has shown two to three-fold increases in PM due to fireworks (Rao et al., 2012;
Ravindra et al., 2003; Tsai et al., 2011; Shen et al., 2009). Greater increases (> 5 times) in
particulate levels elsewhere were related to more intense and prolonged (days) firework activity
(Tian et al., 2014).
Air parcel trajectories arriving at the Manila Observatory during the sampling periods before,
during, and after the firework event were assessed to ascertain the impact of fireworks on the



332 enhanced particulate concentrations. Three-day back trajectories for the period before the
333 firework event were from the northeast to east directions coming from the Philippine Sea (Fig.
334 2a). For the periods (Fig. 2b) during and (Fig. 2c) after the firework event, back trajectories were
335 from the northeast to east/northeast directions. The general wind directions from the back
336 trajectories are consistent with the climatologically prevailing northeasterly monsoonal winds in
337 December and January for the Philippines (Villafuerte II et al., 2014). The origin of the air
338 parcels did not have any major emissions events that could have impacted the measurements
339 after long-range transport. This is important to note because the tracers for fireworks are also
340 tracers for transported emissions due to biomass burning ($K^+$) (Braun et al., 2020) and industrial
341 activities (Cohen et al., 2009). Thus, enriched particulate concentrations during the firework
342 activity were most likely locally produced.

343

344 3.2 Optical Aerosol Properties

345 Heavy aerosol loading at the surface was observed up to eight hours after the fireworks peak (16
346 UTC, 12 AM local time) with high HSRL 532 nm backscatter cross-section and depolarization
347 (Fig. 3a) reaching ~440 m above the ground. Prior to the firework peak, the surface aerosol layer
348 had lower backscatter (before 14 UTC, Fig. 3a), and this cleaner condition is shown by the 08:16
349 UTC vertical profile of the aerosol backscatter (Fig. 3b). Rainfall (Fig. 1a) contributed to
350 columns of high backscatter (Fig. 3a) after 14 UTC and before the firework peak with a
351 measurable decrease in the aerosol backscatter for a short time after the precipitation (15:00 and
352 16:00 UTC).

353 To verify the height values (Fig. 3b), the "surface-attached aerosol layer" height is estimated
354 using the maximum variance method more commonly used for daytime convective boundary
355 layer detection (Hooper and Eloranta, 1986). The method is also limited by the complexity of the
356 case due to pertinent rain signals for this event. The "surface attached aerosol layer" (Fig. 3a) is
357 derived from a 15-min moving window average based on the 30-s values shown with a thin black
358 line. As confirmed by the height detection, aerosols reached up to ~440 m (Fig 3a and b) on 16
359 UTC (31 December 2018). It persisted for at least an hour then dropped to $118 \pm 20$ m with
360 higher aerosol backscatter retained until January 1, 2019 0 UTC. Some of the smoke is above the
361 detected height (i.e. 17 UTC).

362

363 3.3 Mass Size Distributions

364 A total of 41 water-soluble species were detected in the 48-hr size-differentiated particulate
365 samples collected before, during, and after the firework event. The total bulk mass concentration
366 is defined as the sum of the concentrations of all the measured species across the MOUDI's
367 eleven stages ($\geq 0.056$ µm). The total water-soluble bulk mass concentration (Table 1) during the
368 firework event (16.74 µg m$^{-3}$) was 5.71 times and 4.73 times higher than the total bulk mass



concentrations before (2.93 µg m$^{-3}$) and after (3.54 µg m$^{-3}$) the firework event, respectively.
Assuming the average of the water-soluble mass concentrations before and after the firework
event represent background values, this translates to an 80.66% increase in water-soluble mass
during the firework event.
The firework event was associated with increased total water-soluble mass fraction (32.33%)
(0.056 – 3.2 µm size range, Section 3.1) in PM$_{2.5}$ (Fig. S4) compared to before (9.90%) and after
(17.79%) the firework event. The water-soluble particulate mass fraction in PM$_{2.5}$ similarly
increased in other firework events (Yang et al., 2014). The highest total water-soluble mass
concentrations during the firework event were from the following ions: non-sea salt (nss) SO$_4^{2-}$
(6.81 µg m$^{-3}$), K$^+$ (5.05 µg m$^{-3}$), NO$_3^-$ (1.70 µg m$^{-3}$), Cl$^-$ (1.46 µg m$^{-3}$), Mg$^{2+}$ (0.37 µg m$^{-3}$), Na$^+$
(0.33 µg m$^{-3}$), and Ca$^{2+}$ (0.30 µg m$^{-3}$). These contributed to 95.75% of the total detected bulk
water-soluble mass concentration then.
Total water-soluble bulk mass concentration during the firework event was dominated by
submicrometer particles, which accounted for 77.4% of the total water-soluble bulk mass (Fig.
4b). Supermicrometer mass fractions were greater before (Fig. 4a) and after (Fig. 4c) the
firework event (43.7% and 57.5% of the water-soluble bulk mass concentration) compared to
during the firework event (22.6%). The increase in submicrometer mass fractions is typical with
firework emissions (Crespo et al., 2012; Do et al., 2012). In New York, fireworks contributed to
77% of PM$_1$ due to potassium salts and oxidized organic aerosol (Zhang et al., 2019).
Non-sea salt SO$_4^{2-}$ had the highest contribution (40.7%) to total water-soluble bulk mass
concentration during the firework event (Table 1). Sulfate exhibited a shift in its mass size
distribution to a slightly larger size during firework activity (Fig. 4b). During the firework event,
87.13 % of the nss-SO$_4^{2-}$ was in the 0.32 µm to 1.8 µm size fraction. Before and after the
firework event, 87.28% and 85.14% of the nss-SO$_4^{2-}$ mass concentration, respectively, was
distributed in a finer size fraction (0.18 µm to 1 µm) (Fig. 4a and 4c). For context, SO$_4^{2-}$ peaked
at 0.62 µm during fireworks in Nanning, China (Li et al., 2017). Firework emissions include
gases like SO$_2$ which undergo aqueous uptake and oxidation onto particles to form SO$_4^{2-}$.
Furthermore, enhanced secondary formation is aided by metals emitted during fireworks that
help convert SO$_2$ to SO$_4^{2-}$ (Feng et al., 2012; Wang et al., 2007).
Potassium contributed 30.19% to the total water-soluble mass concentration during the firework
event (Table 1), presumably in the form of KNO$_3$. This compound is associated with black
powder used as a propellant (Li et al., 2017). Potassium's mass concentration distribution
similarly shifted to a slightly larger size during the firework event (Figure 4b). Most (87.6%) of
the bulk K$^+$ mass concentration during the firework event was between 0.32 and 1.8 µm,
compared to 85.4% and 79.4% between 0.18 and 1 µm before and after the firework event,
respectively (Fig. 4a and 4c). This is comparable to the mass diameter (0.7 µm) due to firework
emissions after transport in Washington State (Perry, 1999). The shift in the mass size
distribution of K$^+$ and nss-SO$_4^{2-}$ can be due to the removal of nucleation-mode particles as a
result of increased coagulation in the accumulation mode (Zhang et al., 2010).





Nitrate, Cl⁻, and $Mg^{2+}$ mass size distributions all exhibited pronounced peaks in the
submicrometer range during the firework event (Fig. 5). The mass sum concentration of the
aforementioned ions peaked (46.39% of the total mass concentration of the three species)
between 0.56 and 1.0 μm. On the other hand, their mode appeared between 1.8 and 3.2 μm
before and after the firework event (33.02% and 32.91% of the total mass concentration of the
three species, respectively) (Fig. 5). Nitrate, Cl⁻, and $Mg^{2+}$ are emitted during fireworks (Zhang et
al., 2017) as finer-sized submicrometer particles (Tsai et al., 2011) compared to background
conditions when these species are mostly associated with coarser supermicrometer particles
(AzadiAghdam et al., 2019; Cruz et al., 2019; Hilario et al., 2020). Nitrate can also be formed
secondarily (Yang et al., 2014) from firework emissions. Firework emissions are associated with
lower $NO_3^-$:$SO_4^{2-}$ ratios (Feng et al., 2012) compared to days dominated by mobile sources
(Arimoto et al., 1996) due to different formation mechanisms (Tian et al., 2014). Consistent with
the literature, low $NO_3^-$:$SO_4^2$ ratios were also observed during the firework event (before: 0.79,
during: 0.25, after: 0.82). A low $NO_3^-$:$SO_4^{2-}$ ratio is related to decreased pH of the particles (Cao
et al., 2020), which may impact not just air quality and health but also nearby waterbodies where
the particles may deposit. It is important to note that background supermicrometer Cl⁻ and $Mg^{2+}$
in Manila are most likely associated with sea salt while background supermicrometer $NO_3^-$
possibly in the form of $NaNO_3$ (de Leeuw et al., 2001) or $NH_4NO_3$ likely stems from partitioning
of nitric acid gas onto surfaces (de Leeuw et al., 2001) of coarse particles such as sea salt and
dust (AzadiAghdam et al., 2019; Cruz et al., 2019). The Cl⁻:$Na^+$ mass ratio during the firework
event increased to 4.44 (from 0.69 and 1.08 before and after, respectively) and was higher than
the typical Cl⁻:$Na^+$ ratio in seawater of 1.81 (Braun et al., 2017). These ratio results confirm that
the increase in Cl⁻ concentrations during the firework event is not driven by sea salt but instead
linked to firework emissions such as what was shown during Taiwan's lantern festival with Cl⁻
:$Na^+$ ratios reaching approximately 3 owing to raw materials in fireworks such as $KClO_3$, $ClO_3$,
and $ClO_4$ (Tsai et al., 2012). The lack of increased sea salt influence during the firework event,
which is not to be expected, is further confirmed by relatively small changes in the amount of
observed $Na^+$, as will be discussed subsequently.
The $Na^+$, $Ca^{2+}$, and $NH_4^+$ mass size distributions peak in the supermicrometer range (1.8 to 3.2
μm) (Figure S5) and total mass concentrations (Table 1) varied minimally, relative to the earlier
mentioned species, before (0.33 μg m⁻³, 0.21 μg m⁻³, 0.21 μg m⁻³, respectively), during (0.33 μg
m⁻³, 0.30 μg m⁻³, 0.19 μg m⁻³) and after (0.53 μg m⁻³, 0.38 μg m⁻³, 0.28 μg m⁻³) the firework
event. The minimal change in $NH_4^+$ mass concentration is most likely due to little or no variation
of its precursor gas (e.g., $NH_3$) due to firework activities and the fact that firework materials are
commonly composed of K-rich salts rather than $NH_4^+$ salts (Zhang et al., 2019). The latter seems
probable because the K:S mass ratios of 2.75 and 2.71, observed from 0.18 – 0.32 μm and 0.32 –
0.56 μm, respectively, during the firework event suggests a firework-related source of K and S.
This ratio is similar to the K:S ratio of 2.75 (Dutcher et al., 1999) of "black powder" (Perry,
1999), a type of pyrotechnic comprised of K and S.





The mass size distribution for the sum of the rest of the species ("others" in Fig. 4) shifted from
having a peak at the smaller end of the accumulation mode (0.18 – 0.32 µm) before and after the
firework event to larger sizes in the accumulation mode (0.56 – 1.0 µm) during the firework
event. The shift in mode to slightly larger particles during the firework event may be due to
increased coagulation sinks (Zhang et al., 2010) and secondary production (Retama et al., 2019).
An additional coarse peak (3.2 – 5.6 µm) observed after the firework event is mainly attributed
to sea salt constituents (e.g., $Cl^-$, $Na^+$) and likely unrelated to firework emissions aging and
processing. The mass contribution of the "others" to the total measured water-soluble mass
concentration decreased during the firework event to 4.3% from 12.5% before and 11.6% after
the firework event due to the prevalence of the ionic species (nss-$SO_4^{2-}$, $K^+$, $NO_3^-$, $Cl^-$, $Mg^{2+}$, $Na^+$,
$Ca^{2+}$, and $NH_4^+$) discussed earlier (Table 1).

3.4 Enriched Tracers in Firework Emissions
Bulk mass concentrations of eighteen of the forty-one measured species were enriched during the
firework event by more than two times compared to the average of their bulk mass
concentrations before and after the firework event (Fig. 5). Enrichments for Cu (65.2), Sr (24.4),
succinate (19.4), Ba (18.2), $K^+$ (16.3), nss-$SO_4^{2-}$ (9.8), Al (6.9), Pb (6.1), and maleate (5.3) were
highest (> 5) among the species measured (Fig.5). Potassium and nss-$SO_4^{2-}$ together contributed
to 70.9% of the total measured species during the firework event (Table 1). However, Cu, Sr,
succinate, Ba, Al, Pb, and maleate contributed a total of only 2.14% to the total measured species
mass concentration. This reinforces the importance of looking at enrichments rather than
absolute mass concentrations for identifying which aerosol constituents are firework tracers.
Tracer metals in firework emissions were previously shown to contribute a small fraction
(~<2%) to total PM mass (Jiang et al., 2014).
Of the eighteen species with observed enrichments exceeding two (Fig. 5), only those which are
firework components and that are uninfluenced by secondary formation are considered tracers.
The identified fourteen firework tracers based on these criteria are as follows: Cu, Sr, Ba, $K^+$, Al,
Pb, $Mg^{2+}$, Cr, Tl, $Cl^-$, Mn, Rb, Zn, and Ag. Copper gives the blue-violet color of fireworks, Sr
gives the red color, Ba and Tl makes the green flame, and Rb gives a purple color. Potassium and
Ag (as AgCNO or silver fulminate) are propellants, Al is fuel, and Pb provides steady burn and
is also used as an igniter for firework explosions. Chromium is a catalyst for propellants, Mg is a
fuel, and $Mg^{2+}$ is a neutralizer or oxygen donor (U.S. Department of Transportation, 2013).
Manganese is either a fuel or oxidizer, and Zn is used for sparks (Licudine et al., 2012; Martín-
Alberca and García-Ruiz, 2014; Shimizu, 1988; Wang et al., 2007; Ennis and Shanley, 1991).
Metals are usually in the form of $Cl^-$ salts in fireworks (Wang et al., 2007). In this study, the
enrichment of $Cl^-$ during the firework event was found to be 3.7. Some of the identified tracer
metals are regulated and their detection is of concern. Magnesium is not recommended as a
firework component because it is sensitive to heat and can easily ignite in storage (Do et al.,
2012). Lead is highly toxic and thus regulated (Moreno et al., 2010) as its occurrence in





fireworks is not ideal. Although $SO_4^{2-}$, maleate (fuel), and $NO_3^-$ (oxidant) were also enriched
more than two times during the firework event and are also firework components (Zhang et al.,
2019), they can be formed secondarily via gas-to-particle conversion processes (Yang et al.,
2014) and are not considered as firework tracers. Succinate is likewise formed secondarily and is
not considered a firework tracer (Wang et al., 2007).
Size-resolved enrichments (Fig. 5) were highest in the submicrometer range for most measured
species. This is consistent with past studies such as in Italy (Vecchi et al., 2008), Taiwan (Do et
al., 2012), and Spain (Crespo et al., 2012) where elemental concentrations due to pyrotechnics
increased in the submicrometer mode. The peak size differentiated enrichments of the first five
firework tracers Sr (45.08), Ba (57.82), $K^+$ (48.70), Al (18.75), and Pb (69.07) were in the 1.0 –
1.8 µm size range. Copper (49.85) peaked between 0.56 – 1.0 µm because it did not have valid
data for diameters exceeding 1.0 µm. Strontium and Ba had very high enrichments (254.40 and
195.84) from 0.1 – 0.18 µm due to very low concentrations before and after the firework event in
that size range. Enrichments of up to ~1000 (Crespo et al., 2012) for Sr and Ba have been
observed due to pyrotechnics, and both are known firework tracers (Kong et al., 2015).
The size-resolved enrichments of other notable species (Fig. 5 and Fig. S6) peaked at specific
size ranges between 0.32 – 1.8 µm: $Mg^{2+}$ (18.93, 0.056 – 0.1 µm), Cr (14.37, 1.0 – 1.8 µm), Tl
(18.12, 0.56 – 1.0 µm), $Cl^-$ (170.94, 0.32 – 0.56 µm), Mn (6.29, 1.0 – 1.8 µm), Rb (6.87, 1.0 –
1.8 µm), $NO_3^-$ (7.26, 0.56 – 1.0 µm), Cs (6.28, 1.0 – 1.8 µm), Mo (4.15, 0.32 – 0.56 µm), Ti
(6.63, 0.32 – 0.56 µm), Co (17.94, 0.56 – 1.0 µm), and methanesulfonate (MSA) (6.66, 0.56 –
1.0 µm). Among all the measured water-soluble species, $Cl^-$ had the highest size-resolved
enrichment, followed by Sr, Ba, $K^+$, Pb, and Cu.  This is expected because inorganic salts
comprise an enormous percentage of firework emissions (Martín-Alberca et al., 2016).

3.5 SEM-EDX
Five SEM images from the different stages (0.18 – 1 µm) of the MOUDI sampler with possible
firework influence are highlighted (Fig. 7). There were signs of nano-scale aggregation that were
chain-like and reminiscent of soot particles from pyrolysis and combustion (Pirker et al., 2020;
Pósfai et al., 2003; D'Anna, 2015) in all of the images, and especially distinct in the 0.1 – 0.18
µm (Fig. 4b) and 0.18 – 0.32 µm (Fig.7c) stages. Images for larger sizes revealed relatively
larger particles appearing as a translucent crystal-shaped rectangle in the 0.32 – 0.56 µm image
(Fig. 7d), in addition to a capsule-shaped particle (Fig. 7e) and a cubic–shaped particle (Fig. 7f)
in the two 0.56 – 1.0 µm images. The presence of such non-spherical shapes including chain
aggregates points to the potential for particle collapse and shrinking associated with humidified
conditions as noted in past work (Shingler et al., 2016 and references therein).
The chemical composition of the blank Teflon substrate (Fig. 7a) was examined first by EDX to
determine the background signals before the actual samples were analyzed. The color intensity of





the element maps (Fig. S7) relates the concentration of the analyzed element relative to the
backscattered electron image (gray-scale) of the sample. The background substrate was
dominated by C, F, and Al (bright yellow, bright blue, and bright blue-green, respectively, in Fig.
S7-a1/a2/a3). Metallic elements were distributed in each of the five featured SEM images.
Molybdenum and K were present in all of the substrate stages (bright red in Fig. S7-
b3/b4/c3/c8/d7/d8/e6/e7/f6/f9). Other metals were also found in the different stages such as K,
Mg, Al, Ru, Pd, Ba, Hf, and Tl. The identified heavy metals in the particles are commonly used
in firework as fuel components, colorants, and oxidants (Singh et al., 2019). Potassium, Mg, Al,
Ba, and Tl are in the group of firework tracers that were already identified (Section 3.4 and Fig.
5) to have mass bulk concentration enrichments exceeding two. Molybdenum exhibited a
reduced mass bulk concentration enrichment of 1.93 (Fig. 5), but had size-resolved enrichments
between 1.21 and 4.15 (Fig. 6) in the substrate cut-outs analyzed for EDX. The cube-shaped
feature in the 0.56 – 1.0 µm substrate appears to be KCl because of the high color density of K
and Cl in the elemental maps (bright red and bright blue-green in Fig. S7-f6/f8) and because the
shape of KCl is cubic (Pirker et al., 2020). The crystal-shaped rectangle in the 0.32 – 0.56 µm
range appears to be enriched by Cl (bright blue-green in Fig. S7-d6). The same applies to the
capsule-shaped particle in 0.56 – 1.0 µm image (bright blue-green in Fig. S7-e5). The chloride
ion (Cl$^-$) is a component of metal salts, usually in the form of ClO$_4^-$ or ClO$_3^-$ (Tian et al., 2014)
used to color fireworks (Shimizu, 1988).
These results of the sampled portions of the substrate stages are consistent with the results of the
size-resolved submicrometer enrichments measured by IC and ICP-QQQ (Section 3.4) for Mo,
K, Mg, Al, Ba, and Tl. Molybdenum was brightest red in the 0.32 – 0.56 µm image (Fig. S7-d8),
consistent with the highest enrichments (4.15 in Fig. 6) for that size range. Potassium was
brightest red in the 0.56 – 1.0 µm image (Fig. S7-e6/f6), consistent with highest enrichments
(33.04 in Fig. 6). Magnesium was brightest yellow from 0.32 – 1.0 µm (Fig. S7-d4/e3/f4),
consistent with highest enrichments (9.50 and 11.58 in Fig. 6). Aluminum had a high signal in
the blank Teflon substrate but also was brightest blue-green (Fig. S7-d5/e4/f5) in between 0.32 –
1.0 µm in the sample during the firework event, consistent with highest enrichments (9.22 and
13.32 in Fig. 6). Barium was detected by EDX between 0.56 – 1.0 µm (Fig. S7-f11 where its
enrichment was 12.39 (Fig. 6). Thallium was detected between 0.56 and 1.0 µm (Fig. S7-f13) by
EDX, where its enrichment was highest (18.12 in Fig. 6) as detected by ICP-QQQ. The
submicrometer metal salts due to fireworks can uptake water at high humidity (ten Brink et al.,
555     2018).


3.6 Hygroscopicity Analysis
As fireworks alter the chemical profile of ambient PM, we estimate how aerosol hygroscopicity
responded during fireworks relative to periods before and after. For reference, typical κ values
range from 0.1 to 0.5 for diverse air mass types such as urban, marine, biogenic, biomass
burning, and free troposphere (Dusek et al., 2010; Hersey et al., 2013; Shingler et al., 2016;





Shinozuka et al., 2009). AzadiAghdam et al. (2019) reported size-resolved values ranging from
0.02 to 0.31 using data from the same field site in Metro Manila but for a different time period
and without any firework influence (July – December 2018). They found the highest values to be
coincident with MOUDI stages with most sea salt influence (3.2 – 5.6 µm).
For this study, a bulk κ value is reported for the size range between 0.056 – 3.2 µm as noted in
Section 2.6, and subsequent references to composition data are for this size range. Kappa was
enhanced during the firework event (0.18) compared to before (0.11), due mostly to increased
contributions from $K_2SO_4$ and $Mg(NO_3)_2$ (Fig. 8a). More specifically, the volume fractions of
$K_2SO_4$ and $Mg(NO_3)_2$ increased from 0.01 to 0.10 and 0.01 to 0.03, respectively (Fig. 8b). For
context, inorganic salts ($K_2SO_4$, KCl) dominated the aerosol hygroscopicity in Xi'an, China
during fireworks (Wu et al., 2018). In the Netherlands, enhancements in salt mixtures containing
$SO_4^{2-}$, $Cl^-$, $Mg^{2+}$, and $K^+$ were noted to enhance hygroscopicity (ten Brink et al., 2018). Notable
reductions in volume fraction during the firework event were for $NaNO_3$ (0.01 to 0.00), black
carbon (0.26 to 0.12), and $(NH4)_2SO_4$ (0.02 to 0.01) (Fig. 8b). All three species are not
associated with primary firework emissions. Although $NaNO_3$ and $(NH4)_2SO_4$ are hygroscopic,
their decreased volume fractions happened alongside a decreased volume fraction of non-
hygroscopic black carbon and increased volume fractions of the firework-related and
hygroscopic $K_2SO_4$ and $Mg(NO_3)_2$, which increased bulk aerosol hygroscopicity during the
firework event.
Kappa decreased to an intermediate value after the firework event (0.15) (Fig. 8a); this value
exceeds that from before the fireworks owing partly to more sea salt influence that was unrelated
to fireworks. The change in volume fraction of sea salt from before and during fireworks (0.01)
to after the fireworks (0.03) (Fig. 8b) translated to an increase of 0.03 in bulk κ (Fig. 8a) from
before to after the firework event. Although fireworks emit extensive amounts of inorganic
species, the calculated κ values were still relatively low because the background air is dominated
by organics and black carbon, which are relatively hydrophobic species (Table S4) (Cohen et al.,
2009; Oanh et al., 2006; Cruz et al., 2019).

**4. Conclusion**
This study reported on important aerosol characteristics measured during the 2019 New Year
fireworks in Metro Manila. Notable results of this work, following the order of questions raised
at the end of Section 1, are as follows:
• $PM_{2.5}$ was significantly enhanced during firework activities reaching a maximum of
383.9 µg m$^{-3}$ between 01:00 to 02:00 on 1 January 2019. Rainfall, wind, and relative
humidity possibly contributed to washout, local dispersion, and secondary formation of
particles, respectively. A noticeable decrease in aerosol backscatter was measured by the
HSRL lidar for short periods after the rain fall.  There was no significant influence from
long-range transport to the sampling site, confirming that the sample data was most



representative of the local nature of particulate enhancements observed during the
firework event.
• Surface aerosol loading increased over a period of eight hours during the firework event,
coincident with peak PM$_{2.5}$ levels. The heaviest aerosol layer was observed for at least an
hour, and reached ~440 m above the surface, after which the aerosol layer dropped to 118
± 20 m.
• Bulk concentrations of water-soluble species were enhanced especially in the
submicrometer mode during the firework event along with increased water-soluble mass
fractions in PM$_{2.5}$. Potassium and nss-SO$_4^{2-}$ were the major contributors. Mass size
distributions shifted to slightly larger accumulation-mode sizes most likely due to
increased coagulation sinks and secondary formation.
• Components of inorganic salts such as Cu, Sr, Ba, K$^+$, Al, Pb, Mg$^{2+}$, Cr, Tl, Cl$^-$, Mn, Rb,
Zn, and Ag were enriched more than two times during the firework event as compared to
before and after the event. Even while they (excluding K$^+$) comprised only 2.88% of the
total water-soluble mass, their contribution is significant because they support the
findings that the samples represent firework emissions, and especially since some of the
components like Pb and Mg$^{2+}$ are banned substances.
• Cubic and capsule-shaped Cl$^-$-rich particles, suggesting the presence of KCl, were
prominent in submicrometer particles collected during the firework event.
• Aerosol hygroscopicity ($\kappa$) between 0.056 and 3.2 μm increased from 0.11 (before the
fireworks) to 0.18 during the firework event due to the increased volume fractions of
inorganics.

The brief but sharply enhanced concentrations of water-soluble species in the submicrometer size
range, especially for K$^+$ and SO$_4^{2-}$, have implications for both public health and the environment,
the former of which is owing to how smaller particles can penetrate more deeply into the human
respiratory system. Higher concentrations of secondary particles from fireworks are related to
increased mass extinction efficiency and therefore decreased visibility (Jiang et al., 2014). The
increased water-soluble fraction during firework events coincides with elevated particle
hygroscopicity and CCN activity (Drewnick et al., 2006) at smaller diameters (Yuan et al.,
630   2020).


**Data availability**
High Spectral Resolution Lidar data collected at Manila Observatory can be found at:
(University of Wisconsin Lidar Group) http://hsrl.ssec.wisc.edu/by_site/30/custom_rti/
Size-resolved aerosols data collected at Manila Observatory can be found at: (Stahl et al., 2020a)
on figshare as well as on the NASA data repository at
DOI:10.5067/Suborbital/CAMP2EX2018/DATA001.



**Author Contributions**

MTC, MOC, JBS, RAB, ABM, CS, and AS designed the experiments. All coauthors carried out
various aspects of the data collection. MTC, EE, SV, RH, GL, LM, CS, and AS conducted
analysis and interpretation of the data. EE, LM, SV, RH, GL, and AS prepared the manuscript
with contributions from the coauthors.

**Competing Interests**

The authors declare that they have no conflict of interest.

**Acknowledgements**

The authors acknowledge support from NASA grant 80NSSC18K0148 in support of the NASA
CAMP$^2$Ex project. R. A. Braun acknowledges support from the ARCS Foundation. M. T. Cruz
acknowledges support from the Philippine Department of Science and Technology's ASTHRD
Program. A. B. MacDonald acknowledges support from the Mexican National Council for
Science and Technology (CONACYT). We acknowledge Agilent Technologies for their support
and Shane Snyder's laboratories for ICP-QQQ data. We thank the Department of Environment
and Natural Resources Environmental Management Bureau (DENR-EMB) Central Office Air
Quality Management Section in the Philippines and the Air Center for Air Pollution Research in
Japan of EANET for the hourly PM$_{2.5}$ data.

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





**Table 1:** Summary of total and speciated concentrations before, during, and after the firework
event. Species are divided based on units (Total to Zn: µg m$^{-3}$; succinate to Se: ng m$^{-3}$).

| Species | Total Concentration | | | Species | Total Concentration | | |
|---|---|---|---|---|---|---|---|
| | Before | During | After | | Before | During | After |
| TOTAL | 2.93 | 16.74 | 3.54 | MSA | 4.44 | 3.22 | 2.43 |
| nss-SO$_4^{2-}$ | 0.73 | 6.81 | 0.66 | Mn | 0.88 | 2.97 | 1.03 |
| K$^+$ | 0.37 | 5.05 | 0.25 | Rb | 0.62 | 1.24 | 0.25 |
| NO$_3^-$ | 0.64 | 1.70 | 0.65 | Cr | 0.16 | 1.01 | 0.29 |
| Cl$^-$ | 0.23 | 1.46 | 0.57 | As | 0.60 | 0.71 | 0.38 |
| Mg$^{2+}$ | 0.06 | 0.37 | 0.10 | Ni | 0.41 | 0.46 | 0.99 |
| Na$^+$ | 0.33 | 0.33 | 0.53 | Ti | 0.10 | 0.27 | 0.24 |
| Ca$^{2+}$ | 0.21 | 0.30 | 0.38 | V | 0.32 | 0.14 | 0.30 |
| NH$_4^+$ | 0.21 | 0.19 | 0.28 | Mo | 0.05 | 0.10 | 0.06 |
| Ba | 0.01 | 0.17 | 0.01 | Cd | 0.11 | 0.10 | 0.13 |
| oxalate | 0.10 | 0.12 | 0.06 | Co | 0.05 | 0.05 | 0.05 |
| Cu | 2.48E-04 | 6.89E-02 | 1.86E-03 | Cs | 0.02 | 0.02 | 0.01 |
| Al | 4.53E-03 | 0.05 | 0.01 | Ag | 0.02 | 0.02 | 4.00E-04 |
| Sr | 1.27E-03 | 4.65E-02 | 2.54E-03 | Tl | 0.01 | 0.02 | 1.80E-03 |
| Zn | 0.01 | 0.02 | 0.01 | Zr | 0.01 | 0.01 | 0.03 |
| succinate | 0.98 | 9.51 | 0 | Sn | 0.01 | 6.69E-04 | 0.03 |
| Pb | 1.68 | 8.33 | 1.03 | Y | 2.16E-04 | 4.56E-04 | 2.44E-03 |
| phthalate | 12.82 | 5.36 | 5.59 | Nb | 2.28E-04 | 1.59E-04 | 3.00E-04 |
| adipate | 5.35 | 4.83 | 11.73 | Hf | 0 | 0 | 2.18E-04 |
| maleate | 1.54 | 4.12 | 0 | Hg | 1.03E-03 | 0 | 0 |
| Fe | 2.91 | 3.47 | 7.32 | Se | 5.76 | 0 | 0 |


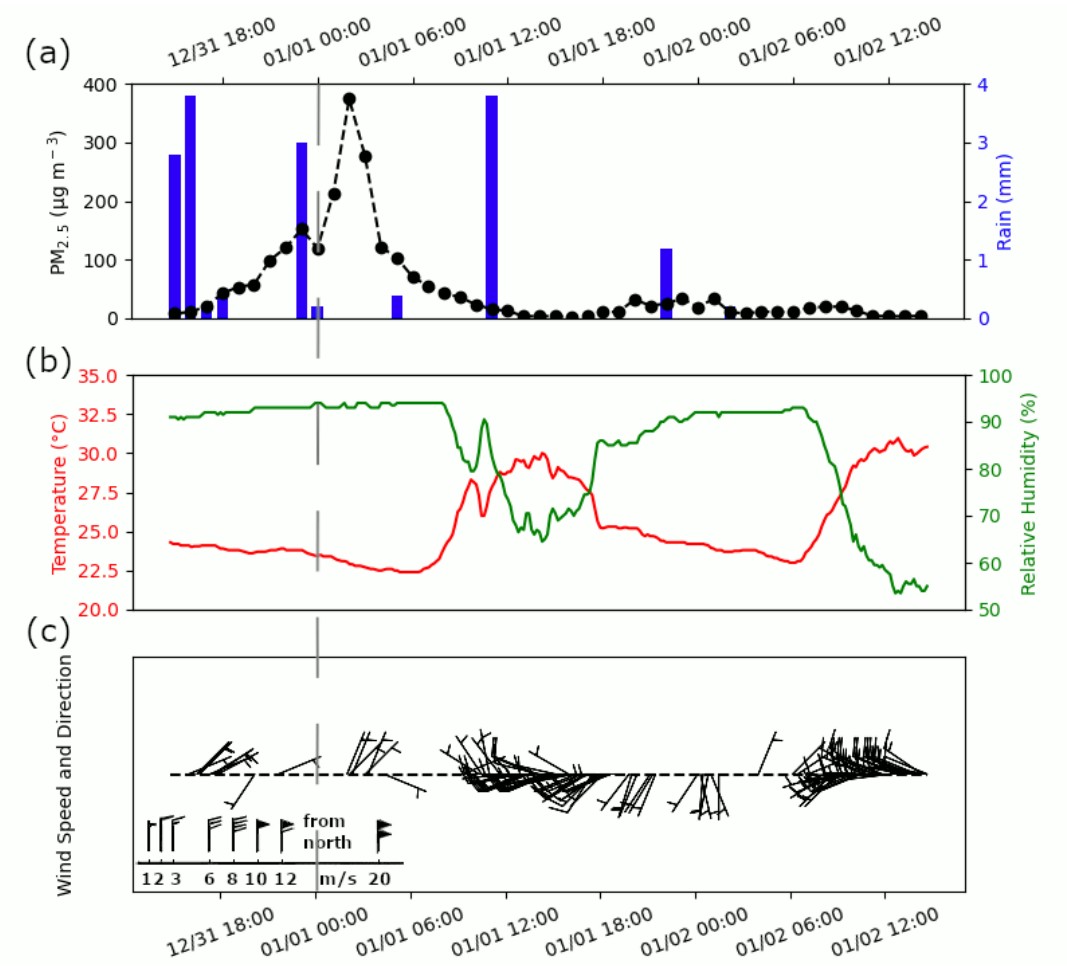


**Figure 1:** (a) PM$_{2.5}$ mass concentrations and rain accumulation at hourly resolution (local time, dashed vertical line indicates midnight) as measured from the Manila Observatory main building third floor rooftop (~88 m.a.s.l.) at the same period as the MOUDI size-speciated samples during the firework event. Ten-minute averaged values of (b) temperature and relative humidity, in addition to (c) wind speed and direction. The wind barb legend in (c) shows how flags are added to the staff with increasing wind speed and in the direction where the wind comes from. Figures S2 and S3 show the hourly PM$_{2.5}$ mass concentrations and ten-minute meteorological data before and after the firework event, respectively.

**Figure 2:** Three-day back trajectories with 6-h resolution for the periods (a) before, (b) during, and (c) after the firework event, ending at the point of the Manila Observatory at 500 m.



**(a)**

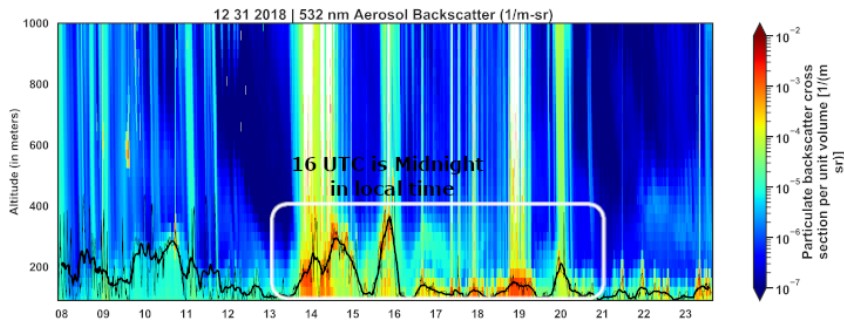

**(b)**

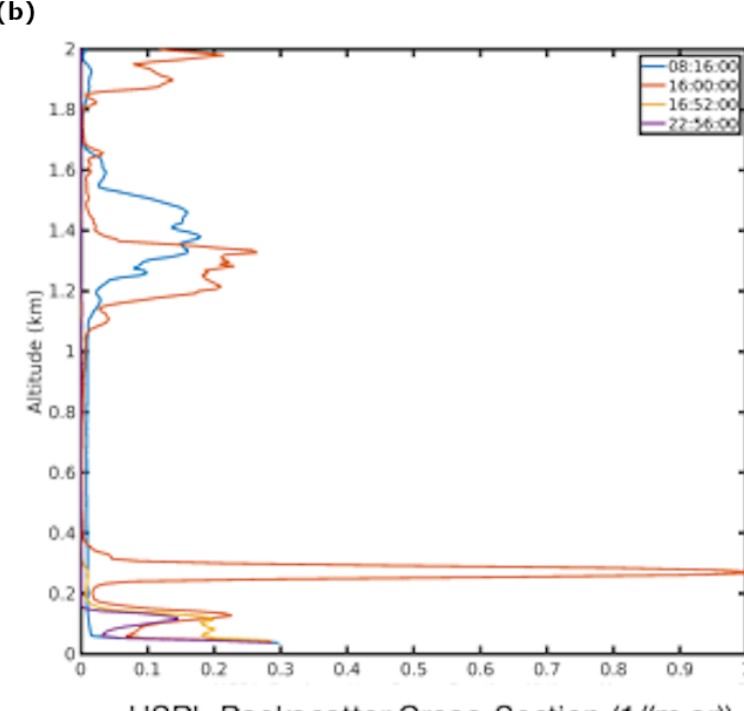


**Figure 3:** (a) Time series of the aerosol backscatter vertical profile from the High Spectral
Resolution Layer (HSRL). The time shown is Universal Time (UT) and local time is UT + 8
hours. The times circled by the white oval correspond to the peak of aerosol backscatter in the
mixing layer due to firework activity. The approximate surface-attached aerosol layer height is
shown as a thick black line. It is derived from a 30-min moving window average based on the 1-
min values shown in thin black line (b) Vertical profiles of aerosol back-scatter at specific UT
times of interest before, during, and after the fireworks.

991                                         29


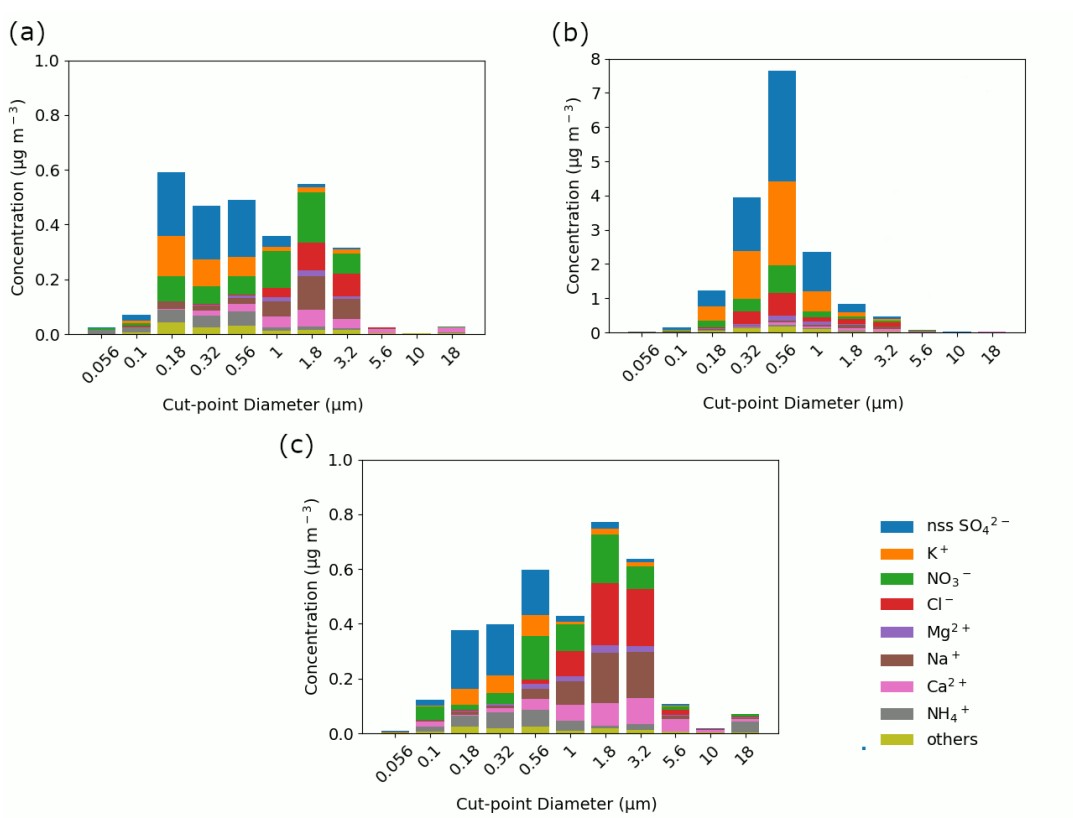

**Figure 4:** Speciated mass size distributions of the major aerosol constituents measured (a) before,
(b) during, and (c) after the firework event. Table 1 lists the bulk (≥ 0.056 μm) mass concentrations
of these ions and elements, including those labeled here as "others" (Ba, oxalate, Cu, Al, Sr, Zn,
succinate, Pb, phthalate, adipate, maleate, Fe, MSA, Mn, Rb, Cr, As, Ni, Ti, V, Mo, Cd, Co, Cs,
Ag, Tl, Zr, Sn, Y, Nb, Hf, Hg, and Se).



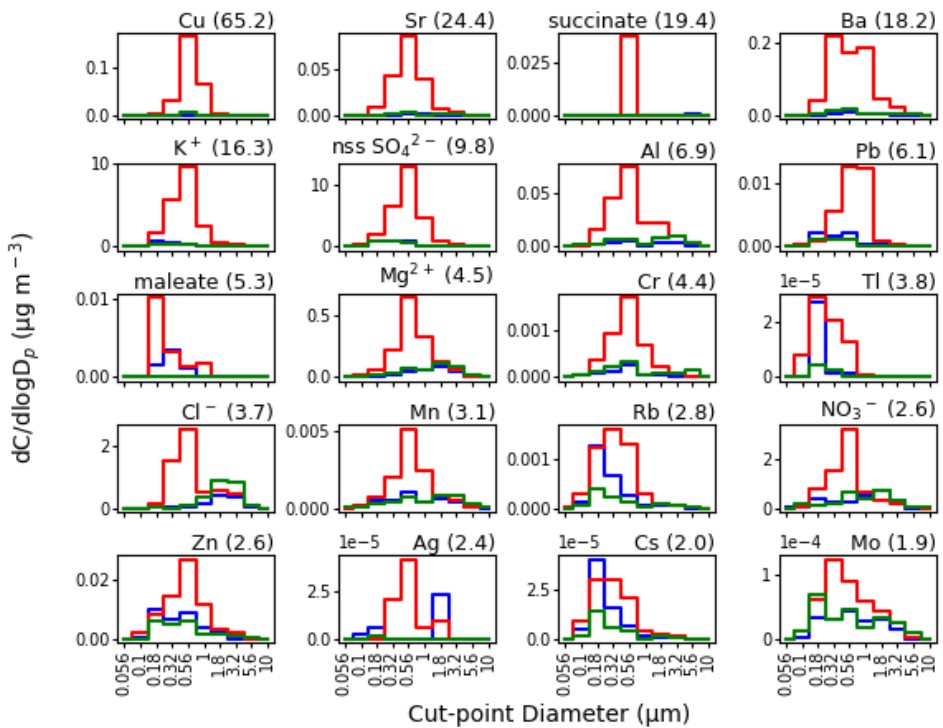

999

**Figure 5:** Speciated mass size distributions before (blue line), during (red line), and after (green line) the firework event. Next to species labels are bulk (≥ 0.056 μm) mass concentration enrichment values due to the firework event; species are shown with enrichments ≥ 1.9. Figure S5 shows similar results for all other species.



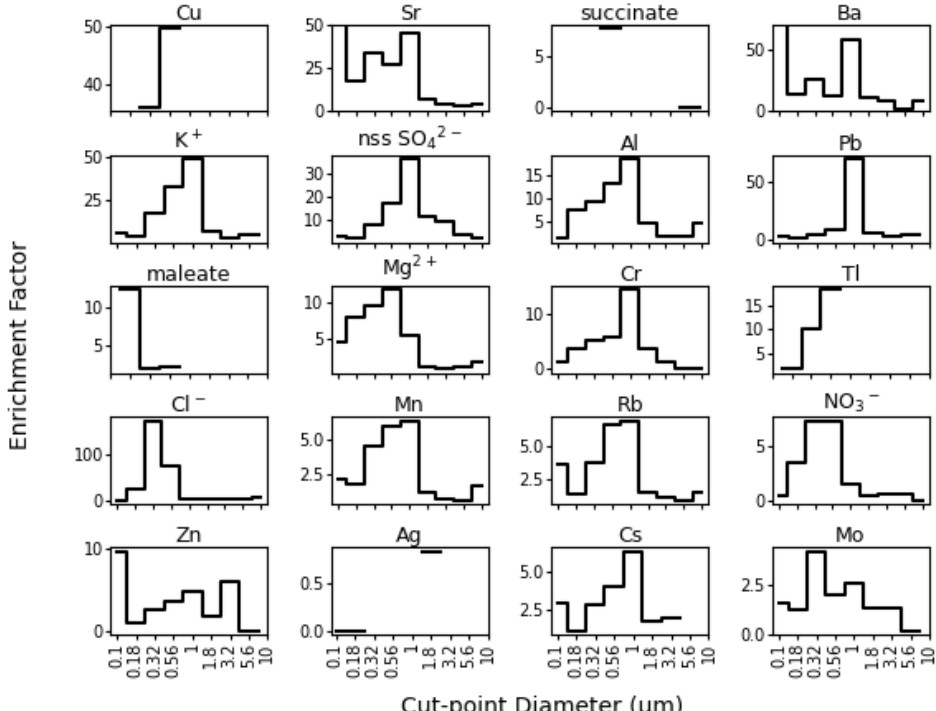

**Figure 6:** Size-resolved enrichments for individual firework tracer species in order of decreasing total bulk mass concentration enrichment (species from Fig. 5). Cut-point diameters with no valid data are left blank. The y-axis of Sr and Ba are truncated to more easily show enrichments in the larger size fractions. Figure S6 shows similar results for all other species.





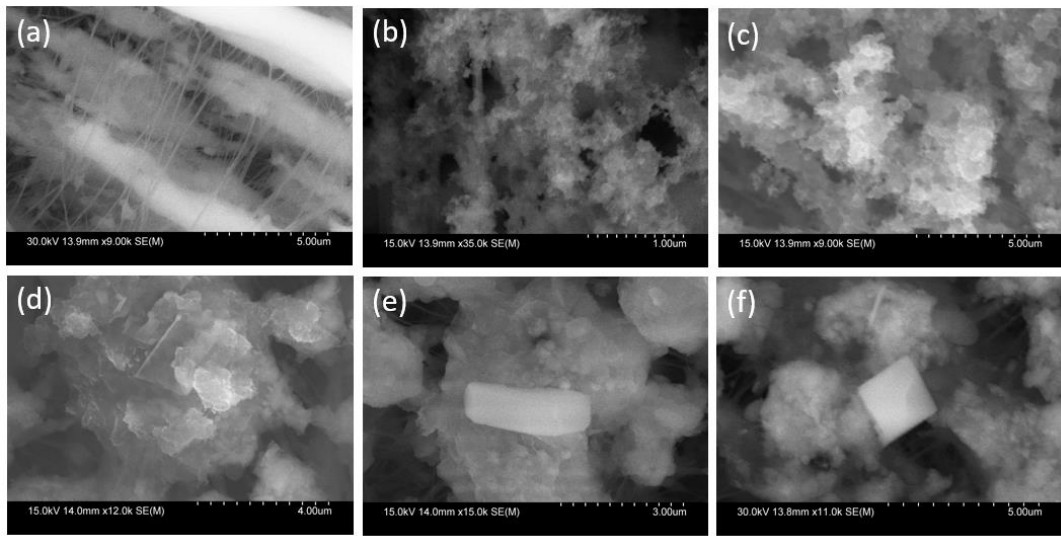

1009

**Figure 7:** Scanning electron microscope (SEM) images of (a) a blank PTFE (Teflon) substrate and (b-f) particles in different diameter ranges with firework influence: (b) 0.1 – 0.18 μm, (c) 0.18 – 0.32 μm, (d) 0.32 – 0.56 μm, (e-f) 0.56 – 1.0 μm.



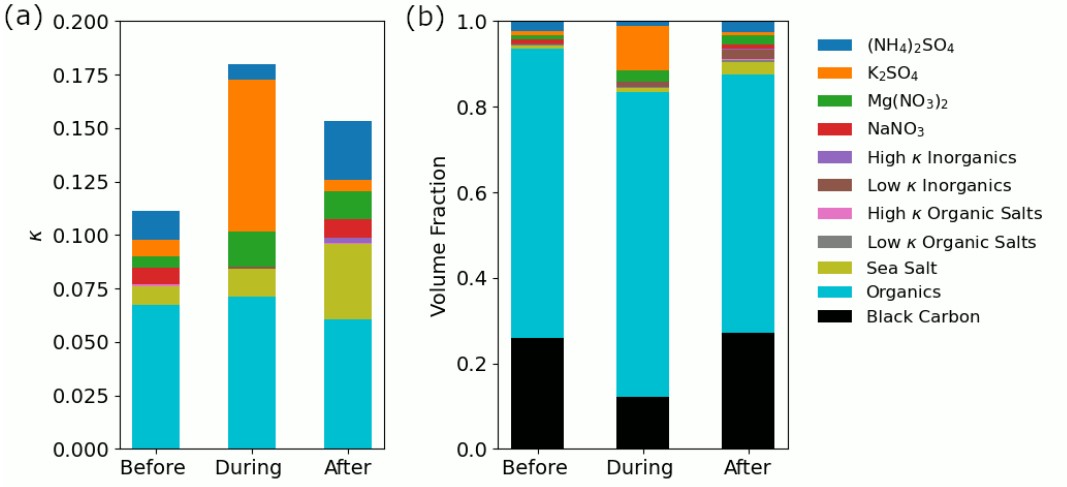

1013

**Figure 8:** (a) Kappa (κ) values for the aerosol fraction between $0.056 - 3.2$ µm before, during, and after the firework event. The speciated contributions to the overall κ values (represented by the colors) are categorized based on the classes of compounds in the legend following past work (AzadiAghdam et al., 2019). Ammonium sulfate, $K_2SO_4$, $Mg(NO_3)_2$, and $NaNO_3$ are high κ inorganics but are plotted separately because of their large contributions. The speciated contributions were calculated by multiplying the volume fraction of each compound class by its intrinsic κ value (Table S4).