# Peer review of "Measurement report: Firework impacts on air quality in Metro Manila, Philippines during the"

_Atmospheric Chemistry and Physics, 2020_

## Referee Comment (RC1) · Anonymous Referee #1 · 25 Nov 2020

In this study, most contents are spent on describing the data without enough discussion. There was no new information on the method development and conclusion. The suggestions are as follows: 1. Fireworks have been widely studied all over the world. Although the studies in the Southeast Asian are not so much, the authors must tell us the difference with other regions and the significance of studying fireworks in this region. 2. There are too many questions that the manuscript wants to address. Please combine some of them, so that the aims of this work can be better understood. 3. Why the carbon fractions were not detected in this work? The manuscript said that "Although fireworks emit extensive amounts of inorganic species, the calculated $\kappa$ values were still relatively low because the background air is dominated by organics and

black carbon, which are relatively hydrophobic species. . .". Carbon fractions accounted for high percentages of PM, and they are important product of fireworks as reported in many literatures. In addition, the carbon aerosol is critical for studying the optical properties and hygroscopicity, which are important parts of this work. Thus, it is a big problem if the carbon fractions were not detected. 4. Many results were reported in this work. However, the explanation and the discussion are lacked. And the relationships among data from different methods must be discussed. 5. The size distribution of chemical compositions can be very useful to study the PM properties, but related discussion is unabundant. And the influence of size distribution of chemical compositions on the optical properties and hygroscopicity must be studied. 6. More evidences (such as fire plots) should be provided and combined to get conclusion. 7. The conclusion should be rewritten. The conclusion now just listed some results of the data. The logical relationship of results must be analyzed and more deeper conclusion must be summarized. 8. The results about compositions have been widely reported, and no new information is provided in this work. The size distribution may be an interesting topic, but it was not studied abundantly in the discussion and no conclusion about it is provided.

---

## Referee Comment (RC2) · Anonymous Referee #2 · 2 Dec 2020

**Statement:**

This article investigates firework pollution during the 2019 New Year celebrations in Manila, Philippines. It takes a comprehensive approach of investigating the emissions and aftermath of the fireworks from a number of angles, including atmospheric composition, meteorological conditions, transport, and growth/decay of particles. It also investigates changes to atmospheric properties as a result of New Year celebrations. There are several measurements taken during this observation period that are unique to this study.

[Figure]

The authors provide an analysis of pollutants, including particulate matter, metals, and toxins. Further, the article includes particle mode analysis. Concentrations of many pollutants, metals and toxins increased dramatically during the celebrations. Some of these dispersed within a few minutes whereas others stayed longer. Some of the observed compounds decreased during the New Year, which is either attributed to interactions with firework emissions or is attributed to the decrease of normal-day human activity, such as traffic. The study also shows that the chemical behavior of the atmosphere, e.g. particle hygroscopicity, can be altered by firework emissions.

Some of the content, especially in the Results and Discussion section, is rather choppy and needs to be restructured. There are numerous comparisons with other cities without much context explained. Some of the content in the Results and Discussion should be moved to the Introduction or Methods sections, noted in the specific comments below.

The results and conclusions of the article include a blend of scientific and detailed technical observations. Consequently, this feels like it is somewhere in between *ACP* and *ACP Measurement Reports.* I would suggest revisions to either make the paper more scientific in nature to submit to ACP, or focus on the new and unique measurements and keep it in *ACP Measurement Reports.*

Should the authors decide to keep the article in *ACP Measurement Reports* with revisions, I would gladly re-review the article.

**Major comments:**

The article feels a bit choppy. It jumps from one subject or result to another without necessarily any coherent transition. A few examples are noted in "Specific comments" below. With some revisions to connect different points together, I think this article would flow much better.

[Figure]

The abstract states, "there have not been any comprehensive physicochemical and optical measurements of fireworks and their associated impacts in a Southeast Asia megacity." A similar statement is made in the Introduction. This statement seems a bit bold and also vague and contradictory to the fact that several other studies of firework celebrations in China and India are cited. Perhaps the authors don't consider China and India to be Southeast Asia, but regardless, this statement needs to be more clear. For example, which measurements have never been done before, and which are new in this study and not in the other cited studies? Is this the first study of its kind in the Philippines?

There are many measurements and results here, and not all of them are linked or compared to each other. This contributes to the choppiness of the paper, and there could be more description of how the different observations and results relate to each other.

The conclusion mostly reiterates the results in bullet point form. This needs to be more concise, with only key findings pointed out. Then the conclusion needs to include more relevance to the aerosol measurement science and/or the greater scientific community and public.

This article was submitted to *ACP Measurement Reports*. In general, there is alignment with the aims of this journal in terms of measurements of various compounds in Manila, which is a new location for such study. This study also contributes new types of measurements. However, the questions in lines 149-155 are more broadly scientific in nature, and the results and conclusions package these results into a more scientific format, similar to other studies on the effects of firework pollution. At the same time, the scientific conclusions are minimal, and focus is very local and not focused on the bigger scientific aims of *ACP*. In its current form, the content and nature of this article feels somewhere in between *ACP* and *ACP Measurement Reports* and not focused on one or the other.

From the website of *ACP Measurement Reports:*

***Measurement reports*** *present substantial new results from measurements of atmospheric properties and processes from field and laboratory experiments. Analysis of the measurements may include model results and conclusions of more limited scope than in research articles.*

Although this study might be the first of its kind in the Philippines, the results are expected and not necessarily new with respect to the many existing publications related to air pollution from firework celebrations. The article needs more emphasis on the aspects of the study that can be considered as "substantial new results."

Therefore, I would suggest the paper be revised as one of the following:

- Revise the overall nature of the paper to focus more on the scientific and societal contribution of the study, and then submit the paper to the main *ACP* journal. In particular, include more in-depth scientific answers to the scientific questions asked in lines 149-155. Additionally, scientific results could, for example, include: How do the results of this study help scientists, policymakers and the general public in not just the Philippines but around the world, and how can these results be used to improve air quality during the New Year in the future?

- Revise to focus more on the aerosol technology, specifically the measurements that are new and unique. There also needs to be more elaboration to how this contributes to the aerosol measurement community. With such revision, this would better align with the aims of *ACP Measurement Reports.*

As an example of what could be revised, the article throws in comparisons with various other cities around the world in with the results/discussion. This shows promise for good scientific content. In its current form, however, these comparisons cause the article to feel choppy. What is the relevance of these comparisons? What do these

comparisons to other cities contribute to either aerosol technology and/or to the general scientific community and public? These comparisons could be elaborated and made more scientific.

One thing that really stood out to me is the toxins, especially lead. This brings to mind a hypothetical question: Could it be possible to use this study to make an argument to policymakers to forbid the use of these toxins or find alternatives to these toxins in fireworks? Although such recommendation might be outside the scope of this specific Measurement Report, elaboration on the seriousness of lead and other toxins in fireworks, which were clearly observed in Manila, could be emphasized more – this could make the paper into a stronger contribution to the scientific community and general public, and it could make the conclusions much stronger.

I would also suggest making a timeseries figure with these metals and toxins, not just a before/during/after figure.

**Specific comments:**

Title: The plural of "Fireworks" plus the second noun "impacts" is not correct English. It should say any of the following: "Firework impacts" or "Impacts of fireworks" or possessive "Fireworks' impacts".

Lines 54-59: Listing these specific numbers from cities around the world is not necessary, and giving these numbers does not add any significance to the article. The two sentences following this are sufficient for this paragraph.

Line 64: "India where" should be "India, and"

Lines 161-163: This sentence doesn't make sense, and it is irrelevant to the article. This topic is not discussed anywhere in the article.

Lines 207-214: The sentence beginning with "Although" through the sentence ending with "study" do not belong in this paragraph. This is introductory material, not methods.

Line 211: The sentence, "There was limited firework after midnight" needs to be more specific and clear – what does "limited" mean, and with respect to what, specifically?

Section 2.7 "Back Trajectories" should be moved to after section 2.2 "Meteorological Data" for better flow of related content and to be consistent with the order in which results are presented.

Line 293-295: This first sentence should be in the introduction or methods section, not results.

Lines 325-329: The last two sentences in this paragraph jump back to talking about fireworks in other countries, which was already stated in the introduction and are now redundant. These two sentences could be deleted. Alternatively, if the intention is to make a scientific comparison of Manila in 2019 to other cities, then this needs to be elaborated, and the comparison needs to be done in more scientific detail.

Lines 330-332: This first sentence was already stated in the methods sections and is redundant here.

Lines 353-358: Again, this is jumping back to methods. Only the last three sentences in this paragraph are the results.

Lines 364-367: These first two sentences are also describing methods of calculation as opposed to results.

Lines 393-394: Again, this jumps back-and-forth from showing results to comparing to another city. If this kind of comparison is desired, then another sentence or two describing the relevance and greater context of this should be added. This should be in a separate paragraph rather than squeezed in the middle of a paragraph reporting numerical results.

Lines 404-405: This is another comparison to a different city that doesn't quite fit in between reporting numerical results from Manila.

Lines 429-433: Here is another comparison to a different location, this time Taiwan. Here, though, the relevance is better explained, and it flows better than these comparisons in other places in the manuscript, but then the following sentence beginning with "The lack of increased sea salt" jumps back to results/discussion in Manila. I would suggest the comparison to Taiwan be moved to a separate paragraph.

Lines 485-486: "Lead is highly toxic and thus regulated (Moreno et al., 2010) as its occurrence in fireworks is not ideal." – I would say it's more than just "not ideal;" it sounds like a serious health hazard to me.

Lines 570-573: This is again a place where the text jumps into comparisons with specific other cities. The relevance and context needs to be elaborated a bit more.

Figure 3: Why does this figure use UTC when the other figures use local time? Then there is unnecessary text in the middle of (a) stating that 16UTC is midnight local time. I would suggest using local time because the study is with respect to the New Year (centered around 00:00) and to be consistent with other figures.

---

## Author Comment (AC1) · 14 Jan 2021

Response: We thank the two reviewers for thoughtful suggestions and constructive criticism that have helped us improve our manuscript. Below we provide responses to reviewer concerns and suggestions in blue font. All changes to the manuscript can be identified in the version submitted using Track Changes.

Anonymous Referee #1

In this study, most contents are spent on describing the data without enough discussion. There was no new information on the method development and conclusion. The

suggestions are as follows:

Response: Thank you for your blunt evaluation but we want to also remind the reviewer that according to ACPD, a Measurement Report aims to do the following: "Measurement reports present substantial new results from measurements of atmospheric properties and processes from field and laboratory experiments. Analysis of the measurements may include model results and conclusions of more limited scope than in research articles." We fit into this category as we report substantial new results from field measurements of aerosols over Metro Manila (satisfying the first sentence in the quotation above). Further, we discuss the results and reach a series of conclusions (satisfying the second sentence in the quotation above). If there is any concern that the conclusions are of limited scope, that is fine and not an issue as that is partly the nature of Measurement Reports based on the quotation above.

1. Fireworks have been widely studied all over the world. Although the studies in the Southeast Asian are not so much, the authors must tell us the difference with other regions and the significance of studying fireworks in this region.

Response: Thank you for this comment. China (East Asia) and India (South Asia) where most of the other firework studies have been done are not part of Southeast Asia (SEA) that includes Cambodia, Laos, Myanmar (Burma), Peninsular Malaysia, Thailand, Vietnam, Brunei, East Malaysia, East Timor, Indonesia, Philippines, Singapore, and a small part of India. East Asia and South Asia have different meteorological conditions and geology as compared to SEA. SEA has a unique hydrometeorological condition (high moisture) and geology (islands and mainland) and as such complicates the study of aerosols in the area. Aerosol studies in SEA are generally limited. An extreme event such as New Year with fireworks adds to the complexity and there are scarce studies on this. More specifically, there are no size-speciated chemical analysis as well as optical properties of firework emissions in Manila. Results of this work can improve understanding of the local impacts on health and the environment, which currently are still lacking. Several sentences and sources were added to the introduction

about these. Here is the added text:

"Studies on the properties of aerosols in general in South East Asia (Tsay et al., 2013) which is one of the rapidly developing regions in Asia are limited. This compounds the challenge to understand the interactions between aerosols and the complex hydro-meteorological and geological environment in South East Asia (Reid et al., 2013). Increased local and transported emissions (Hopke et al., 2008; Oanh et al., 2006) in South East Asia adds to the complexity and affects air quality in the region. Firework emissions are an example of extreme and regular local emissions in South East Asia. And even while several studies exist in the neighboring regions of East Asia and South Asia, there currently is no in-depth analysis of the chemical, physical, and optical properties of firework emissions in a South East Asian megacity where fireworks are culturally significant. Studies on the impacts on health and the general environment due to firework emissions in South East Asia are as scarce."

2. There are too many questions that the manuscript wants to address. Please combine some of them, so that the aims of this work can be better understood.

Response: Thank you for this note, we have condensed the questions to two major questions: "We address the following questions in order: (i) what are the conditions of the atmosphere during the study period in relation to aerosols, and how are these affected by firework emissions?; and (ii) what are the concentrations, mass size distributions, and morphological characteristics of different elemental and ionic species specific to fireworks, and how do these affect bulk aerosol hygroscopicity?"

3. Why the carbon fractions were not detected in this work? The manuscript said that "Although fireworks emit extensive amounts of inorganic species, the calculated $\kappa$ values were still relatively low because the background air is dominated by organics and black carbon, which are relatively hydrophobic species: : :". Carbon fractions accounted for high percentages of PM, and they are important product of fireworks as reported in many literatures. In addition, the carbon aerosol is critical for studying the

optical properties and hygroscopicity, which are important parts of this work. Thus, it is a big problem if the carbon fractions were not detected.

Response: We were not able to analyze both for elemental and organic carbon because of the need for separate analysis which was not available at that time. We used related literature on a study done in a nearby site to estimate the ratio of elemental carbon in the samples. An issue was insufficient substrate surface area to do all of the various types of analyses possible. We cut the substrates into portions for the different types of analyses we report in the paper and there was insufficient sample left for more detailed carbon analysis.

4. Many results were reported in this work. However, the explanation and the discussion are lacked. And the relationships among data from different methods must be discussed.

Response: Thanks for this note, hopefully the condensed conclusions and rearranged sentences helped to address the connection between the data, analysis, and implications.

5. The size distribution of chemical compositions can be very useful to study the PM properties, but related discussion is unabundant. And the influence of size distribution of chemical compositions on the optical properties and hygroscopicity must be studied.

Response: We have added text to address the implications of the composition results for optical and hygroscopic properties:

"Higher concentrations of secondary particles, which in this study is in the accumulation mode, from fireworks are related to increased mass extinction efficiency and therefore decreased visibility (Jiang et al., 2014) as was observed. The increased water-soluble fraction, especially in the submicrometer mode, during firework events coincides with elevated particle hygroscopicity which is related to CCN activity (Drewnick et al., 2006) at smaller diameters (Yuan et al., 2020) and which can be part of a future study."

6. More evidences (such as fire plots) should be provided and combined to get conclusion.

Response: We are not fully sure what the reviewer is referring to here as the suggestion is vague to us. We feel that we have reported our measurement data effectively and comprehensively already. We have tried to improve our conclusions, as also mentioned in response to the next comment. Extending the analysis is beyond the scope of this work as this is just a Measurement Report where we report special data for a specific region; more extensive analysis and discussion would warrant a regular article submission which is not what our intention is at this point.

7. The conclusion should be rewritten. The conclusion now just listed some results of the data. The logical relationship of results must be analyzed and more deeper conclusion must be summarized.

Response: Thank you for this note. We have rewritten the conclusion to address the two major science questions in the introduction. We reordered the sentences also to give the context of the work and connect the analysis and results better and more fluidly.

8. The results about compositions have been widely reported, and no new information is provided in this work. The size distribution may be an interesting topic, but it was not studied abundantly in the discussion and no conclusion about it is provided.

Response: We believe the novelty of this work is the "combination" of different datasets used to characterize firework emissions in a critically important (and highly populated area) without a detailed firework study in the peer-reviewed literature yet. We respectfully disagree that we did not study the size distribution behavior as that is the foundation of our analysis (i.e., the MOUDI data). We have addressed this comment thought by adding some more discussion and improved conclusions.

2020-1028, 2020.

Drewnick, F., Hings, S. S., Curtius, J., Eerdekens, G., and Williams, J.: Measurement of fine particulate and gas-phase species during the New Year's fireworks 2005 in Mainz, Germany, Atmospheric Environment, 40, 4316-4327, 2006. Hopke, P. K., Cohen, D. D., Begum, B. A., Biswas, S. K., Ni, B., Pandit, G. G., Santoso, M., Chung, Y.-S., Davy, P., and Markwitz, A.: Urban air quality in the Asian region, Science of the Total Environment, 404, 103-112, 2008. Jiang, Q., Sun, Y., Wang, Z., and Yin, Y.: Aerosol composition and sources during the Chinese Spring Festival: fireworks, secondary aerosol, and holiday effects, ACPD, 14, 20617-20646, 2014. Oanh, N. K., Upadhyay, N., Zhuang, Y.-H., Hao, Z.-P., Murthy, D., Lestari, P., Villarin, J., Chengchua, K., Co, H., and Dung, N.: Particulate air pollution in six Asian cities: Spatial and temporal distributions, and associated sources, Atmospheric environment, 40, 3367-3380, 2006. Reid, J. S., Hyer, E. J., Johnson, R. S., Holben, B. N., Yokelson, R. J., Zhang, J., Campbell, J. R., Christopher, S. A., Di Girolamo, L., and Giglio, L.: Observing and understanding the Southeast Asian aerosol system by remote sensing: An initial review and analysis for the Seven Southeast Asian Studies (7SEAS) program, Atmospheric Research, 122, 403-468, 2013. Tsay, S.-C., Hsu, N. C., Lau, W. K.-M., Li, C., Gabriel, P. M., Ji, Q., Holben, B. N., Welton, E. J., Nguyen, A. X., and Janjai, S.: From BASE-ASIA toward 7-SEAS: A satellite-surface perspective of boreal spring biomass-burning aerosols and clouds in Southeast Asia, Atmospheric environment, 78, 20-34, 2013. Yuan, L., Zhang, X., Feng, M., Liu, X., Che, Y., Xu, H., Schaefer, K., Wang, S., and Zhou, Y.: Size-resolved hygroscopic behaviour and mixing state of submicron aerosols in a megacity of the Sichuan Basin during pollution and fireworks episodes, Atmospheric Environment, 226, 117393, 2020.

---

## Author Comment (AC2) · 14 Jan 2021

Response: We thank the two reviewers for thoughtful suggestions and constructive criticism that have helped us improve our manuscript. Below we provide responses to reviewer concerns and suggestions in blue font. All changes to the manuscript can be identified in the version submitted using Track Changes.

Anonymous Referee #2

Statement: This article investigates firework pollution during the 2019 New Year celebrations in Manila, Philippines. It takes a comprehensive approach of investigating

the emissions and aftermath of the fireworks from a number of angles, including atmospheric composition, meteorological conditions, transport, and growth/decay of particles. It also investigates changes to atmospheric properties as a result of New Year celebrations. There are several measurements taken during this observation period that are unique to this study.

The authors provide an analysis of pollutants, including particulate matter, metals, and toxins. Further, the article includes particle mode analysis. Concentrations of many pollutants, metals and toxins increased dramatically during the celebrations. Some of these dispersed within a few minutes whereas others stayed longer. Some of the observed compounds decreased during the New Year, which is either attributed to interactions with firework emissions or is attributed to the decrease of normal-day human activity, such as traffic. The study also shows that the chemical behavior of the atmosphere, e.g. particle hygroscopicity, can be altered by firework emissions. Some of the content, especially in the Results and Discussion section, is rather choppy and needs to be restructured. There are numerous comparisons with other cities without much context explained. Some of the content in the Results and Discussion should be moved to the Introduction or Methods sections, noted in the specific comments below.

The results and conclusions of the article include a blend of scientific and detailed technical observations. Consequently, this feels like it is somewhere in between ACP and ACP Measurement Reports. I would suggest revisions to either make the paper more scientific in nature to submit to ACP, or focus on the new and unique measurements and keep it in ACP Measurement Reports. Should the authors decide to keep the article in ACP Measurement Reports with revisions, I would gladly re-review the article.

Response: Thank you for the thoughtful feedback. We have kept the paper as a Measurement Report and made the necessary revisions to address both reviewer comments, and hope this version is viewed as improved by this reviewer.

Major comments: The article feels a bit choppy. It jumps from one subject or result to another without necessarily any coherent transition. A few examples are noted in "Specific comments" below. With some revisions to connect different points together, I think this article would flow much better.

The abstract states, "there have not been any comprehensive physicochemical and optical measurements of fireworks and their associated impacts in a Southeast Asia megacity." A similar statement is made in the Introduction. This statement seems a bit bold and also vague and contradictory to the fact that several other studies of firework celebrations in China and India are cited. Perhaps the authors don't consider China and India to be Southeast Asia, but regardless, this statement needs to be more clear. For example, which measurements have never been done before, and which are new in this study and not in the other cited studies? Is this the first study of its kind in the Philippines?

Response: Thank you for this comment. Yes, China (East Asia) and India (South Asia) are not part of South East Asia (Cambodia, Laos, Myanmar (Burma), Peninsular Malaysia, Thailand, Vietnam, Brunei, East Malaysia, East Timor, Indonesia, Philippines, Singapore, and small part of India). We add text to clarify studies in Southeast Asia (SEA) are limited (this is the first to our knowledge for the Philippines in the peer-reviewed literature) and that India and China are not considered part of SEA: "Studies on the properties of aerosols in general in South East Asia (Tsay et al., 2013) which is one of the rapidly developing regions in Asia are limited. This compounds the challenge to understand the interactions between aerosols and the complex hydro-meteorological and geological environment in South East Asia (Reid et al., 2013). Increased local and transported emissions (Hopke et al., 2008; Oanh et al., 2006) in South East Asia adds to the complexity and affects air quality in the region. Firework emissions are an example of extreme and regular local emissions in South East Asia. And even while several studies exist in the neighboring regions of East Asia and South Asia, there currently is no in-depth analysis of the chemical, physical, and optical properties of firework

emissions in a South East Asian megacity where fireworks are culturally significant. Studies on the impacts on health and the general environment due to firework emissions in South East Asia are as scarce." We also clarify what is unique about our study in terms of our technical approach. We specifically use a wide blend of datasets which are not commonly used altogether to study fireworks, including size-resolved aerosol measurements (e.g.. ionic/elemental composition, morphology), HSRL-2, PM2.5 and meteorology). The following text was added: "And even while several studies exist in the neighboring regions of East Asia and South Asia, there currently is no in-depth analysis of the chemical, physical, and optical properties of firework emissions in a South East Asian megacity where fireworks are culturally significant. This study is novel because it includes, for the first time aerosol data during fireworks, including size-resolved measurements (e.g. ionic/elemental composition, morphology), HSRL-2, PM2.5 and meteorology."

There are many measurements and results here, and not all of them are linked or compared to each other. This contributes to the choppiness of the paper, and there could be more description of how the different observations and results relate to each other.

Response: We try to reduce the so-called choppiness by adding more transition statements between the different types of analyses we present. We also try to harmonize the results and observations better, especially in the conclusions. Here are examples of transition sentences we have now in the draft:

"We begin with hourly PM2.5 mass concentration results for the study period to provide context for the spatio-temporal characteristics of fine particulates due to fireworks, their interaction with meteorology, and effects on aerosol optical properties."

"One factor impacting surface PM concentrations is the vertical structure of the lower troposphere, which is addressed in the next section based on HSRL data."

"Building on the previous results describing the general environmental conditions during the study period, now we focus on the detailed size-resolved measurements. "

"Here we more closely examine how much concentrations of species changed during the firework event. "

The conclusion mostly reiterates the results in bullet point form. This needs to be more concise, with only key findings pointed out. Then the conclusion needs to include more relevance to the aerosol measurement science and/or the greater scientific community and public.

Response: We revised conclusions and it has fewer bullets. We tried to make it more concise with only the most important findings. We also tried to emphasize its relevance to broader themes.

This article was submitted to ACP Measurement Reports. In general, there is alignment with the aims of this journal in terms of measurements of various compounds in Manila, which is a new location for such study. This study also contributes new types of measurements. However, the questions in lines 149-155 are more broadly scientific in nature, and the results and conclusions package these results into a more scientific format, similar to other studies on the effects of firework pollution. At the same time, the scientific conclusions are minimal, and focus is very local and not focused on the bigger scientific aims of ACP. In its current form, the content and nature of this article feels somewhere in between ACP and ACP Measurement Reports and not focused on one or the other.

From the website of ACP Measurement Reports: Measurement reports present substantial new results from measurements of atmospheric properties and processes from field and laboratory experiments. Analysis of the measurements may include model results and conclusions of more limited scope than in research articles.

Although this study might be the first of its kind in the Philippines, the results are expected and not necessarily new with respect to the many existing publications related

to air pollution from firework celebrations. The article needs more emphasis on the aspects of the study that can be considered as "substantial new results."

Therefore, I would suggest the paper be revised as one of the following:

• Revise the overall nature of the paper to focus more on the scientific and societal contribution of the study, and then submit the paper to the main ACP journal. In particular, include more in-depth scientific answers to the scientific questions asked in lines 149-155.

Additionally, scientific results could, for example, include: How do the results of this study help scientists, policymakers and the general public in not just the Philippines but around the world, and how can these results be used to improve air quality during the New Year in the future?

• Revise to focus more on the aerosol technology, specifically the measurements that are new and unique. There also needs to be more elaboration to how this contributes to the aerosol measurement community. With such revision, this would better align with the aims of ACP Measurement Reports.

Response: We respond to the string of suggestions above all at once here because the string relates to the same theme of whether this paper is a Measurement Report or not. We originally intended for it to be a Measurement Report and still stand by this idea with the submitted draft. We break down each of the 2 sentences from the ACP website about what a Measurement Report is and we justify why our previous version and the revised version fit into this category:

"Measurement reports present substantial new results from measurements of atmospheric properties and processes from field and laboratory experiments.": We indeed present new and valuable data and results about atmospheric properties from field measurements. There should be no question about this hopefully.

"Analysis of the measurements may include model results and conclusions of more limited scope than in research articles.": We analyze the measurement data and present results and conclusions. They may arguably be more limited in scope that research articles because they may be mostly specific to the Philippines region. But again, the limitation of this study having been done in one site is why we originally even considered that this would eventually be placed into a Measurement Report category. We don't feel (like the reviewer said) that we need especially high focus on "aerosol technology" as we aren't focused on a methods/instrument paper (otherwise that would be a AMT submission). If we put in too much discussion and comparisons with other regions, we do not feel that hurts the paper but instead makes it stronger, especially for a Measurement Report type of paper.

As an example of what could be revised, the article throws in comparisons with various other cities around the world in with the results/discussion. This shows promise for good scientific content. In its current form, however, these comparisons cause the article to feel choppy. What is the relevance of these comparisons? What do these comparisons to other cities contribute to either aerosol technology and/or to the general scientific community and public? These comparisons could be elaborated and made more scientific.

Response: We have addressed these by moving some of the comparisons to the Introduction for better flow and background information before getting into the results. Examples of the text now are as follows:

"In Nanning, China, $SO_4^{2-}$ peaked at 0.62 $\mu$m during fireworks (Li et al., 2017). The mass diameter of $K^+$ was 0.7 $\mu$m due to firework emissions after transport in Washington State (Perry, 1999)." This sentence was moved from the discussion of results and now appears in the introduction section in the paragraph on size distributions.

"Inorganic salts ($K_2SO_4$, $KCl$) dominated the aerosol hygroscopicity in Xi'an, China during fireworks (Wu et al., 2018). In the Netherlands, enhancements in salt mixtures containing $SO_4^{2-}$, $Cl^-$, $Mg^{2+}$, and $K^+$ were noted to enhance hygroscopicity (ten Brink

et al., 2018)." This sentence was also removed from the discussion of the results and moved to the introduction section on composition.

One thing that really stood out to me is the toxins, especially lead. This brings to mind a hypothetical question: Could it be possible to use this study to make an argument to policymakers to forbid the use of these toxins or find alternatives to these toxins in fireworks? Although such recommendation might be outside the scope of this specific Measurement Report, elaboration on the seriousness of lead and other toxins in fireworks, which were clearly observed in Manila, could be emphasized more – this could make the paper into a stronger contribution to the scientific community and general public, and it could make the conclusions much stronger.

Response: Excellent suggestion, thank you. We have included a phrase on the hazardous effect of Pb to health in the conclusion. We also add mention to a very recently published paper on lead in the Metro Manila region (Gonzalez et al., 2021). Here is our added text:

"The presence of Pb in the firework emissions exacerbates the presence of submicrometer Pb in Metro Manila (Gonzalez et al., 2021)."

I would also suggest making a timeseries figure with these metals and toxins, not just a before/during/after figure.

Response: This is currently not possible because we only have three data points in time (accumulated sample of 2 days for before, during and after) for each metal and toxin. We hope we understood what you meant. Had we obtained more data at better than 2 day time resolution, this would have been an excellent addition.

Specific comments: Title: The plural of "Fireworks" plus the second noun "impacts" is not correct English. It should say any of the following: "Firework impacts" or "Impacts of fireworks" or possessive "Fireworks' impacts".

Response: Thank you, we removed the "s" tailing Firework. Now the title reads...

"Firework impacts"...

Lines 54-59: Listing these specific numbers from cities around the world is not necessary, and giving these numbers does not add any significance to the article. The two sentences following this are sufficient for this paragraph.

Response: We removed the specific numbers from the text but kept the list of cities and then combined that with the following sentence:

"Total PM mass concentrations during local celebrations in different cities: Leipzig, Germany, (Wehner et al., 2000), Texas, United States [U.S.], (Karnae, 2005), Montreal, Canada (Joly et al., 2010), and New Delhi, India, (Mönkkönen et al., 2004) exceeded the 24 h U.S. National Ambient Air Quality Standard (NAAQS) for PM10 of 150 $\mu$g m-3."

Line 64: "India where" should be "India, and"

Response: We replaced to "India, and"...

Lines 161-163: This sentence doesn't make sense, and it is irrelevant to the article. This topic is not discussed anywhere in the article.

Response: Ok, we deleted that sentence.

Lines 207-214: The sentence beginning with "Although" through the sentence ending with "study" do not belong in this paragraph. This is introductory material, not methods.

Response: We needed to include that sentence because locally there is also firework activity on December 24 which is included in the date of the background sample (before) used for the New Year firework sampling (December 31 to January 2). We included the dates for the before, after, and during samples in the sentence before for context.

Line 211: The sentence, "There was limited firework after midnight" needs to be more specific and clear – what does "limited" mean, and with respect to what, specifically?

Response: We changed the preceding sentence:

"Firework activity around the sampling site began around ∼19:00 on December 31, 2018, peaked at 00:00 of 1 January 2019, and dropped drastically after. Based on PM2.5 data there was no evidence of sustained firework activity past midnight."

Section 2.7 "Back Trajectories" should be moved to after section 2.2 "Meteorological Data" for better flow of related content and to be consistent with the order in which results are presented.

Response: Thank you for this note, we reordered the section and moved Back Trajectories to section 2.3 and reordered the sections that came after. We also edited in-line text that may have been affected by this reordering.

Line 293-295: This first sentence should be in the introduction or methods section, not results.

Response: Thank you for this, we deleted this first sentence in the results and added the following to the first sentence of the Methods section "the evolution of and the":

"Hourly PM2.5 mass concentrations were obtained to assess the evolution of and the temporal characteristics of fine particulates due to fireworks and their relation to meteorology and aerosol optical properties."

Lines 325-329: The last two sentences in this paragraph jump back to talking about fireworks in other countries, which was already stated in the introduction and are now redundant. These two sentences could be deleted. Alternatively, if the intention is to make a scientific comparison of Manila in 2019 to other cities, then this needs to be elaborated, and the comparison needs to be done in more scientific detail.

Response: The last two sentences were revised. The first sentence was revised to emphasize that the results are comparable to past studies, and that greater increases (second sentence) have been observed where there were more firework activity in general (Chinese New Year, more intense and prolonged, lasting several days)). The

edited two sentences follow:

"Two to three-fold increases in PM mass concentration due to fireworks has also been observed in previous work in other countries (Rao et al., 2012; Ravindra et al., 2003; Tsai et al., 2011; Shen et al., 2009). Greater increases (> 5 times) in particulate mass concentrations elsewhere were related to more intense and prolonged "(lasting several days)" firework activity (Tian et al., 2014)"

Lines 330-332: This first sentence was already stated in the methods sections and is redundant here.

Response: We removed this first sentence from the results and added the following to the methods section for context: "To ascertain the impact of fireworks on the surface particulate concentrations,…"

Lines 353-358: Again, this is jumping back to methods. Only the last three sentences in this paragraph are the results.

Response: We moved these sentences to the methods section and edited appropriately. Here is what it looks like in the methods section:

"To verify the height values based on the vertical profiles of aerosol backscatter, the "surface-attached aerosol layer" height is estimated using the maximum variance method more commonly used for daytime convective boundary layer detection (Hooper and Eloranta, 1986). The height detection method is limited by the complexity of the firework event case due, however, to pertinent rain signals. The "surface attached aerosol layer" is derived from a 15-min moving window average based on the 30-s values."

Lines 364-367: These first two sentences are also describing methods of calculation as opposed to results.

Response: We moved these sentences to the methods section (2.5 Aerosol Composition and Morphology Measurements).

Lines 393-394: Again, this jumps back-and-forth from showing results to comparing to another city. If this kind of comparison is desired, then another sentence or two describing the relevance and greater context of this should be added. This should be in a separate paragraph rather than squeezed in the middle of a paragraph reporting numerical results.

Response: We were doing the comparison in order to suggest possible mechanisms for the slightly larger sulfate particle size during fireworks. The Li article makes a suggestion that it is because sulfate is formed secondarily during the fireworks and that particle aging contributes also to growth. We moved the information about the size to the introduction. Then we moved the other sentences in the noted line numbers to another paragraph after the discussion of K+ results.

Lines 404-405: This is another comparison to a different city that doesn't quite fit in between reporting numerical results from Manila.

Response: We moved the size info of the different city to the introduction.

Lines 429-433: Here is another comparison to a different location, this time Taiwan. Here, though, the relevance is better explained, and it flows better than these comparisons in other places in the manuscript, but then the following sentence beginning with "The lack of increased sea salt" jumps back to results/discussion in Manila. I would suggest the comparison to Taiwan be moved to a separate paragraph.

Response: Thank you for this note, and for the appreciation. . . we moved the note on the Taiwan results to the introduction as there was only one sentence about this and would have been insufficient for another paragraph.

Lines 485-486: "Lead is highly toxic and thus regulated (Moreno et al., 2010) as its occurrence in fireworks is not ideal." – I would say it's more than just "not ideal;" it sounds like a serious health hazard to me.

Response: We have changed the wording to "serious health hazard"

Lines 570-573: This is again a place where the text jumps into comparisons with specific other cities. The relevance and context needs to be elaborated a bit more. Figure 3: Why does this figure use UTC when the other figures use local time? Then there is unnecessary text in the middle of (a) stating that 16UTC is midnight local time. I would suggest using local time because the study is with respect to the New Year (centered around 00:00) and to be consistent with other figures.

Response: Thank you, we moved the note on the different countries to the introduction. We changed the time units to local time instead of UTC for Figure 3.

[revised manuscript text omitted]

---

## Referee Report (RR1)

Referee Re-Review: "Measurement report: Firework impacts on air quality in Metro Manila, Philippines during the 1 2019 New Year revelry"
Anonymous Referee #2
February 1, 2021

**Statement:**

This manuscript has presented new measurements of air quality in Manila, Philippines during the 2019 new year. Many toxins and hazardous air quality measurements were observed to be enhanced during this time. The manuscript and its results showed great promise. There were many observations, and there was certainly not a lack of content. Some of these measurements are novel and have never yet been done in a Southeast Asian city.

The biggest concern I had with the initial submission of the manuscript was that it felt rather disorganized. In particular, different sections were not linked together, there weren't very well-described relationships between the sections, and there didn't seem to be clear or coherent connections between them. In the results section, there were a number of comparisons to other cities around the world that felt somewhat unclear and perhaps out of place. Moreover, manuscript tried to answer too many scientific questions, rather than focusing on the scope of the measurements, as described in the mission of *Atmospheric Chemistry and Physics: Measurement Reports.*

The authors responded with an Author Comment along with submitting a new version, which I believe has addressed all my concerns. There is now much better flow and consistency between sections. The results are presented much more clearly. The authors have also simplified their research questions down to two main questions they want to address, which are now stated clearly in the introduction. Consistent with these two research questions, the conclusion has been simplified to directly answer them.

I suggested timeseries figures for the metals, but the authors have clarified that measurements were made only at a few points in time, and thus they have presented the best available data.

In the revised submission, I noticed five minor technical/typographical issues, noted in the comments below. With pleasure, I would recommend to the Editor that this manuscript be published in *Atmospheric Chemistry and Physics: Measurement Reports*, once these specific issues are addressed.

36 **Specific comments:**

38 Line 164: There is a reference to "PSA, 2015", but this does not appear to be in the references.

40 Line 166: There is a question mark immediately followed by a semicolon. Just one or the other should be
41 used (either would work).

43 Line 236: Standard convention is "UTC" not "UT"

45 Lines 243-245: The statement, "Although there is some firework activity that is expected in the evening
46 of December 24 (before the firework event), this is minimal compared to that which is the focus of this
47 study" should have a reference.

49 Lines 488-495, which describe the uses of metals in fireworks including which metal gives each color, is
50 introductory material and should be moved to the section starting at line 76. Same with the two sentences
51 about magnesium (lines 497-500). Actually, it seems most of these statements are redundant. For
52 example, "Sr gives the red color" is said in both places, and therefore the second time can be removed.

---

## Author Response (AR2)

Response: We thank the editor for the decision to publish subject to minor revisions. These revisions have
helped improve our manuscript. Below we provide responses to the editor notes and suggestions in blue
font. All changes to the manuscript can be identified in the version submitted using Track Changes.

Editor Decision: Publish subject to minor revisions (review by editor) (12 Feb 2021)

Comments to the Author:

Dear authors,

Thanks for the very good responses to the referee comments.

Please make the suggested additional edits provided by the referees.

We have already edited the manuscript as suggested by the referees and the details are found in the
individual responses to them.

Please also

1) Provide the figures on white background
All figures are on white background.
2) Use the same font size and fontweight in the texts in the different sub-plots a), b) etc in diffierent
figures.

The font size and font-weight have been adjusted in the text of the sub-plots of Figure 3.

[Figure]

3) Please provide higher resolution figures as several of them suffer from low-resolution .jpg conversion.

The resolution of the figure in Figure 3(b) has been improved.

yours,

-tuukka

Response: We thank the two reviewers for thoughtful suggestions and constructive criticism that have
helped us to continue to improve our manuscript. Below we provide responses to referee re-review notes
and suggestions in blue font. All changes to the manuscript can be identified in the version submitted
using Track Changes.

Referee Re-Review: "Measurement report: Firework impacts on air quality in Metro Manila, Philippines
during the 1 2019 New Year revelry"
Anonymous Referee #1
February 8, 2021

The manuscript has been improved. There are some technical errors.  Please check the manuscript
carefully.

Thank you very much for the encouragement and the notes.  We have re-checked the manuscript and have
noted down the responses we have for the specific issues mentioned.

For example the quality of Figure 3 is poor;

The quality of Figure 3, specifically Figure 3(b), has been improved.

The legend should be added in Figure 5.

A legend has been added in the upper left portion of Figure 5.

In line 589 and Fig. 8(b), I can't understand why it is volume fraction?

We added the following to the text to remind the reader about how bulk kappa is calculated based on
individual kappa's and volume fractions of the compounds:

"This is expected because based on the ZSR mixing rule (Stokes and Robinson, 1966) the bulk
hygroscopicity ($\kappa$) is dependent on the sum of the $\kappa$ values for individual non-interacting compounds
weighted by their respective volume fractions."

We also added the letter "(b)" to the Figure 8 description:

"The speciated contributions were calculated by multiplying the (b) volume fraction of each compound
class by its intrinsic $\kappa$ value (Table S4)."

Referee Re-Review: "Measurement report: Firework impacts on air quality in Metro Manila, Philippines
during the 1 2019 New Year revelry"
Anonymous Referee #2
February 1, 2021
**Statement:**

This manuscript has presented new measurements of air quality in Manila, Philippines during the 2019
New Year. Many toxins and hazardous air quality measurements were observed to be enhanced during
this time. The manuscript and its results showed great promise. There were many observations, and there
was certainly not a lack of content. Some of these measurements are novel and have never yet been done
in a Southeast Asian city.

The biggest concern I had with the initial submission of the manuscript was that it felt rather
disorganized. In particular, different sections were not linked together, there weren't very well-described
relationships between the sections, and there didn't seem to be clear or coherent connections between
them. In the results section, there were a number of comparisons to other cities around the world that felt
somewhat unclear and perhaps out of place. Moreover, manuscript tried to answer too many scientific
questions, rather than focusing on the scope of the measurements, as described in the mission of
*Atmospheric Chemistry and Physics: Measurement Reports.*

The authors responded with an Author Comment along with submitting a new version, which I believe
has addressed all my concerns. There is now much better flow and consistency between sections. The
results are presented much more clearly. The authors have also simplified their research questions down
to two main questions they want to address, which are now stated clearly in the introduction. Consistent
with these two research questions, the conclusion has been simplified to directly answer them.

I suggested time series figures for the metals, but the authors have clarified that measurements were made
only at a few points in time, and thus they have presented the best available data.

In the revised submission, I noticed five minor technical/typographical issues, noted in the comments
below. With pleasure, I would recommend to the Editor that this manuscript be published in *Atmospheric*
*Chemistry and Physics: Measurement Reports*, once these specific issues are addressed.
**Specific comments:**
Line 164: There is a reference to "PSA, 2015", but this does not appear to be in the references.

The following reference was added.
"PSA: NCR Statistics: http://rssoncr.psa.gov.ph/, access: February 13, 2021, 2015."

Line 166: There is a question mark immediately followed by a semicolon. Just one or the other should be
used (either would work).

Only the semicolon was retained.

Line 236: Standard convention is "UTC" not "UT"

This was changed to "UTC".

Lines 243-245: The statement, "Although there is some firework activity that is expected in the evening
of December 24 (before the firework event), this is minimal compared to that which is the focus of this
study" should have a reference.

We added references about the firework culture in the Philippines and also previous New Year data from
the government institution.

"Dela Piedra, M. C.: A Filipino Tradition: The Role of Fireworks and Firecrackers in the Philippine
Culture, TALA, 1, 141-153, 2018.

Roca, J. B., de Los Reyes, V. C., Racelis, S., Deveraturda, I., Sucaldito, M. N., Tayag, E., and O'Reilly,
M.: Fireworks-related injury surveillance in the Philippines: trends in 2010–2014, Western Pacific
surveillance and response journal: WPSAR, 6, 1, 2015.

Santos Flora, L., Pabroa, C. B., Morco, R. P., and Racho, J. M. D.: Elemental characterization of
inhalable particulate emissions on New Year's day in Metro Manila, Philippines Nuclear Journal, 15, 35-
43, 2010."

Lines 488-495, which describe the uses of metals in fireworks including which metal gives each color, is
introductory material and should be moved to the section starting at line 76. Same with the two sentences
about magnesium (lines 497-500). Actually, it seems most of these statements are redundant. For
example, "Sr gives the red color" is said in both places, and therefore the second time can be removed.

The description of the metals from Thallium (and so on) were removed from the results and transferred to
the introduction as suggested. The text inserted in the introduction is below. The redundant statements
were also removed.

"Thallium makes a green flame. Potassium and Ag (as AgCNO or silver fulminate) are propellants, Al is
fuel, and Pb provides steady burn and is also used as an igniter for firework explosions. Chromium is a
catalyst for propellants, Mg is a fuel, and $Mg^{2+}$ is a neutralizer or oxygen donor (U.S. Department of
Transportation, 2013). Manganese is either a fuel or oxidizer, and Zn is used for sparks (Licudine et al.,
2012; Martín-Alberca and García-Ruiz, 2014; Shimizu, 1988; Wang et al., 2007; Ennis and Shanley,
1991)."

---

## Author Response (AR3)

Response: We thank the editor for the decision to publish subject to technical corrections. Below we
provide responses to the editor notes and suggestions in blue font. All changes to the manuscript can be
identified in the version submitted using Track Changes.

Editor Decision: Publish subject to technical corrections (review by editor) (19 Feb 2021)

Comments to the Author:

Dear authors,

For some reason I still observe a light green background in Figures 1, 2, 4, and 8.

We ensured a white background in the imaging software (GIMP), and resaved as *.png before inserting in
the word document.  The figures are below. Multiple co-authors have checked and didn't see the problem.
We also tried to view the file as *.pdf to check as well. Hopefully this something that can be addressed by
the journal office during copy-editing.

[Figure]

**Figure 1:** (a) PM$_{2.5}$ mass concentrations and rain accumulation at hourly resolution (local time, dashed vertical line indicates midnight) as measured from the Manila Observatory main building third floor rooftop (~88 m.a.s.l.) at the same period as the MOUDI size-speciated samples during the firework event. Ten-minute averaged values of (b) temperature and relative humidity, in addition to (c) wind speed and direction. The wind barb legend in (c) shows how flags are added to the staff with increasing wind speed and in the direction where the wind comes from. Figures S2 and S3 show the hourly PM$_{2.5}$ mass concentrations and ten-minute meteorological data before and after the firework event, respectively.

[Figure]

**Figure 2:** Three-day back trajectories with 6-h resolution for the periods (a) before, (b) during, and (c) after the firework event, ending at the point of the Manila Observatory at 500 m.

[Figure]

**Figure 4:** Speciated mass size distributions of the major aerosol constituents measured (a) before, (b)
during, and (c) after the firework event. Table 1 lists the bulk ($\geq$ 0.056 μm) mass concentrations of these
ions and elements, including those labeled here as "others" (Ba, oxalate, Cu, Al, Sr, Zn, succinate, Pb,
phthalate, adipate, maleate, Fe, MSA, Mn, Rb, Cr, As, Ni, Ti, V, Mo, Cd, Co, Cs, Ag, Tl, Zr, Sn, Y, Nb,
Hf, Hg, and Se).

[Figure]

**Figure 8:** (a) Kappa (κ) values for the aerosol fraction between 0.056 – 3.2 µm before, during, and after the firework event. The speciated contributions to the overall κ values (represented by the colors) are categorized based on the classes of compounds in the legend following past work (AzadiAghdam et al., 2019). Ammonium sulfate, $K_2SO_4$, $Mg(NO_3)_2$, and $NaNO_3$ are high κ inorganics but are plotted separately because of their large contributions. The speciated contributions were calculated by multiplying the (b) volume fraction of each compound class by its intrinsic κ value (Table S4).

The resolution is poor in Figures 5 and 6.

We have increased the resolution of Figures 5 and 6. The improved figures are found below.

[Figure]

**Figure 5:** Speciated mass size distributions before (blue line), during (red line), and after (green line) the firework event. Next to species labels are bulk (≥ 0.056 μm) mass concentration enrichment values due to the firework event; species are shown with enrichments ≥ 1.9. Figure S5 shows similar results for all other species.

[Figure]

**Figure 6:** Size-resolved enrichments for individual firework tracer species in order of decreasing total bulk mass concentration enrichment (species from Fig. 5). Cut-point diameters with no valid data are left blank. The y-axis of Sr and Ba are truncated to more easily show enrichments in the larger size fractions. Figure S6 shows similar results for all other species.